**TOOLS**

# Design principles of human membrane protein topology

Haoxi Wu[1] and Ramanujan S. Hegde[2]

**We have curated and annotated the topologic determinants for all human membrane proteins made at the endoplasmic reticulum. This census of 4,863 proteins allowed us to systematically analyze the physical properties of their 20,546 transmembrane domains (TMDs) and flanking soluble regions. Single-pass proteins house the majority of large exoplasmic and cytosolic domains, whereas multipass proteins overwhelmingly contain short loops and tails. All classes of TMDs have positively charged cytosolic flanks, but negatively charged exoplasmic flanks feature primarily on TMDs inserted by Oxa1 family insertases. The TMD pair, a topologic unit of two TMDs with a short exoplasmic loop, is the dominant building block of multipass proteins. TMD pairs accommodate high-hydrophilicity and charge-containing TMDs crucial for multipass protein functions. We interpret these context-dependent TMD features in light of current mechanistic models for membrane protein biogenesis and function. Our findings have implications for the evolution of membrane proteomes and for engineering new membrane proteins.**

## Introduction

The human genome is thought to encode ~5,000 integral membrane proteins (Krogh et al., 2001). Membrane proteins function broadly in all aspects of life, including pathogen detection, neurotransmission, nutrient uptake, ion transport, signaling, and many more (von Heijne, 2007). A salient feature of membrane proteins is their transmembrane domains (TMDs), either alpha-helices or beta-barrels. While all beta-barrel proteins are inserted into the outer mitochondrial membrane and a subset of alpha-helical proteins are inserted into the inner and outer mitochondrial membranes, the vast majority of alpha-helical membrane proteins are initially inserted into the membrane of the endoplasmic reticulum (ER), which will be the focus of this article.

Membrane proteins are targeted to the ER by one of two qualitatively different mechanisms: cotranslationally or posttranslationally (Pool, 2022). The latter route is used primarily by ~220 tail-anchored (TA) proteins, which contain a single TMD within ~50 amino acids (aa) of the C terminus (Hegde and Keenan, 2011; Guna et al., 2023). This TMD serves as the targeting sequence and sole membrane anchor. The remainder are targeted cotranslationally via the signal recognition particle (SRP) (Akopian et al., 2013). SRP typically recognizes the first hydrophobic domain that emerges from the ribosome and delivers the translating ribosome to a receptor at the ER membrane.

For some membrane proteins, the targeting element is a cleavable N-terminal signal sequence (SS) (Liaci and Förster, 2021), which directs the downstream soluble domain across the ER to enforce an $N_{exo}$ topology (i.e., with the N terminus facing the noncytosolic, or exoplasmic, side of the membrane). The remainder of membrane proteins are targeted to the membrane using their first TMD, often termed a signal anchor (SA) because it serves as both a SS and a membrane anchor (Spiess and Lodish, 1986). By this definition, the TMD of a TA protein is also an SA, although its mechanism of targeting is qualitatively different (Hegde and Keenan, 2011). An SA can be inserted into the membrane in either topology: $N_{exo}$ or $N_{cyt}$ (i.e., N terminus facing the cytosol) (Spiess et al., 2019; Higy et al., 2004).

At the ER, proteins are primarily inserted into the membrane by one of two universally conserved routes: through the Sec61 protein–conducting channel or with the aid of an Oxa1 family insertase (Rapoport et al., 2017; McDowell et al., 2021; Hegde and Keenan, 2022). Emerging evidence over the past few years indicates that Sec61-mediated insertion is for TMDs whose translocated flanking domain is long (typically >100 aa), and hence requires a channel to cross the membrane. TMDs whose flanking domain is shorter than ~50 aa use an Oxa1 family member, which is thought to locally thin the membrane to facilitate short-domain translocation (McDowell et al., 2021; Hegde and Keenan, 2024; Chen et al., 2017). TMDs with intermediate flanking lengths are probably capable of accessing either route to some degree. The mammalian ER contains three Oxa1 family members:

[1]Department of Biochemistry, University of Oxford, Oxford, UK;   [2]MRC Laboratory of Molecular Biology, Cambridge, UK.

Correspondence to Haoxi Wu: haoxi.wu@bioch.ox.ac.uk;   Ramanujan S. Hegde: rhegde@mrc-lmb.cam.ac.uk.

GET, used exclusively for TA proteins; EMC, used for TA and non-TA proteins; and GEL, probably used only for multipass membrane proteins (McDowell et al., 2021; Hegde and Keenan, 2022).

Although the molecular mechanisms are understood only partially, it is clear that the biophysical properties and positional context of TMDs direct their route of targeting, pathway of insertion, and final topology relative to the membrane (Hegde and Keenan, 2024). Because topology determines which regions of a protein face the cytosol and which face the exoplasmic side of the membrane, it is a critical parameter of many aspects of membrane protein biology. First, many protein modifications are compartment-specific, such as glycosylation in the ER lumen (Aebi, 2013) or palmitoylation in the cytosol (Charollais and Van Der Goot, 2009). Second, trafficking signals, such as for ER export (Barlowe, 2003) or endocytosis (Bonifacino and Traub, 2003), need to face the correct side of the membrane to be recognized. Third, facing one side or the other necessarily determines the set of potential protein interaction partners, and hence protein fate and function (von Heijne, 2006).

These and other considerations mean that knowledge of a protein's topology can immediately constrain models of various aspects of its biology such as trafficking, modifications, interaction partners, and potential functions. For example, many candidate interaction partners found in high-throughput experiments can be excluded because they are not topologically plausible. Similarly, motifs for glycosylation, autophagy adaptors, phosphorylation, and others can be meaningful or not depending on whether they face the appropriate side of the membrane. Hence, accurate predictions of protein topology are crucial for rationalizing and interpreting existing data and predicting protein function.

Two major parameters are used to identify TMDs and predict their topology, mostly considering TMDs in isolation. The first is hydrophobicity, and the second is TMD flanking charged residues. TMDs are generally hydrophobic helix–favoring sequences of ∼16–25 aa (Kyte and Doolittle, 1982; Hessa et al., 2005; Hessa et al., 2007). The ∼5–15 aa of TMD flanking sequence that faces the cytosol (i.e., the "inside" of the cell) is generally positively charged, the so-called positive-inside rule (von Heijne, 1989; von Heijne, 1986; Hartmann et al., 1989). These considerations, together with the fact that each TMD has the opposite topology of the preceding one, can be used to infer protein topology. While these general trends are informative, there are many exceptions that make accurate predictions challenging (Ott and Lingappa, 2002).

One problem is that TMDs of multipass proteins can be very hydrophilic, overlapping substantially with hydrophobic stretches of soluble proteins (Hessa et al., 2007). They can also be unusually long or short, or contain helix-disfavoring aa such as proline and glycine (De Marothy and Elofsson, 2015). These TMDs often cannot insert into or reside stably in the membrane when tested in isolation (Hessa et al., 2005; Hessa et al., 2007; Hedin et al., 2010). Yet, in the context of a correctly folded protein, they are stabilized by other TMDs that pack together to expose a belt of hydrophobicity toward the membrane while burying hydrophilic regions in the protein's interior (Smalinskaitė and Hegde, 2022; Cymer et al., 2015). Similarly, the positive-inside rule can have exceptions for individual TMDs, but holds well in the overall context (von Heijne, 2006).

Given the importance of protein topology for function, and the overall importance of membrane proteins in general, we have sought to curate accurate topology models for all human membrane proteins that are made at the ER. We combine classical topology prediction algorithms (Hallgren et al., 2022, *Preprint*; Tsirigos et al., 2015) with recent genome-wide protein structure predictions (Tunyasuvunakool et al., 2021), together with manual inspection, to generate a census of human membrane protein topology. The results are made accessible to the nonexpert via a topology viewer web interface, with details downloadable for further analysis by the community.

We then use our curated and annotated dataset to determine systematic patterns of topologic and biophysical properties of the human membrane proteome. The findings reveal how a TMD's context within a protein is intimately correlated to the biophysical features that are tolerated by that TMD. By matching these patterns to current knowledge about the routes and mechanisms of Sec61-mediated and Oxa1 family–mediated TMD insertion, we provide insight into the preferences and limits of these universally conserved insertion machineries.

## Results

### Curation of the human membrane proteome

As outlined in Fig. S1, we began with 5,548 potential human membrane proteins that were predicted by at least one previous database as having one or more alpha-helical TMDs. The primary sources were the AlphaFold-based AlphaFold Transmembrane protein (AFTM) database (Pei and Cong, 2023) and UniProt (UniProt Consortium, 2025), both of which draw from multiple experimental and prediction-based sources for their annotations. Mitochondrial proteins (324 as annotated on UniProt and further cross-checked with MitoCarta 3.0 [Rath et al., 2021]) were removed. Of the remainder, the number of predicted TMDs for each protein across databases (Lomize et al., 2017; Dobson et al., 2023; Pei and Cong, 2023; UniProt Consortium, 2025; Dobson et al., 2015; Kozma et al., 2013) was assessed to identify 3,307 with full concordance, and 1,917 with conflicts. Of the concordant proteins, 6 obsolete entries were removed, and another 7 were removed after manual inspection of the evidence suggested mitochondrial or cytosolic localization.

The 1,917 conflicted proteins were each manually inspected by several criteria to judge the evidence for TMD(s) and localization. In addition to available experimental data in structural databases and published work, candidate TMDs were assessed by a combination of AlphaFold structural predictions (Jumper et al., 2021; Tunyasuvunakool et al., 2021), transmembrane helix prediction (TMHMM) profiles (Hallgren et al., 2022, *Preprint*), and full protein scans with the "biological hydrophobicity scale" to identify segments with the lowest free energies (ΔG) for membrane insertion (Hessa et al., 2005; Hessa et al., 2007). In the AlphaFold structures, we assessed potential TMDs by helical prediction of at least 15 aa, a (mostly) hydrophobic surface, and, where applicable, helical bundles with a membrane-thick belt of primarily hydrophobic character. These criteria led to the elimination of 45 likely secreted proteins, 3 obsolete entries, 10

likely hairpin proteins that do not fully traverse the membrane, and 290 non-TMD proteins (mostly cytosolic).

For the remaining 4,863 proteins, N-terminal cleavable signal peptides were taken from UniProt annotations, or, where unclear, from SignalP predictions (Teufel et al., 2022). TMD boundaries were derived from AlphaFold predictions and experimentally determined structures (PDB) using the OPM server (Lomize et al., 2012). The boundaries of extramembrane segments (referred to as "loops" when between two TMDs and "tails" for N- or C-terminal segments) were deduced from TMD boundaries. The orientation of each protein was curated manually based on existing experimental data, domains known to face one side or the other, and homology to proteins of known topology. In cases where such information was not available, we inspected the surface hydrophobicity and charge of AlphaFold-predicted or experimental structures and deduced the likely topology according to the "positive-inside rule" relative to the likely membrane plane.

The fully curated set of membrane proteins with sequences, topologic information, and annotated TMD and non-TMD regions is available as Table S1. To make this information accessible and easily browsable, a web-based viewer was implemented (https://topology.bioch.ox.ac.uk/). For any protein (or set of proteins), the user can see basic topologic information, a topology diagram, and a link to the UniProt page (Fig. S2). All parameters and sequence elements can be downloaded as a CSV file, and the diagram can be downloaded as an SVG file. This resource will facilitate systematic bioinformatics analyses of human membrane proteins to deduce their core design and biogenesis principles, the focus of the remainder of this work.

## Topologic features of the membrane proteome

Of the membrane proteins produced at the human ER, 46% (2,248) contain a single TMD and 54% (2,615) contain multiple TMDs (Fig. 1 A). Among single-pass proteins, most (1,247) begin with a cleavable SS and are necessarily in the $N_{exo}$ topology (also called Type I membrane proteins). The remaining single-pass proteins (1,001) contain an SA in either the $N_{cyt}$ topology with a long C-terminal tail (477, also called Type II membrane proteins), $N_{exo}$ topology (304, also called Type III membrane proteins), or $N_{cyt}$ topology with a short C-terminal tail (220, TA proteins, also called Type IV membrane proteins).

Unlike single-pass proteins, only 9% of multipass proteins (247) begin with a cleavable SS (and these are all $N_{exo}$), with the remaining 91% (2,368) using a SA for ER targeting. Similar to single-pass proteins, multipass proteins that begin with a SA have a roughly 3:2 ratio of $N_{cyt}$ to $N_{exo}$ topology (1,390 and 978, respectively). The number of TMDs among multipass proteins is highly nonuniform in distribution, with certain values being strongly favored in a topology-dependent manner. Among $N_{exo}$ multipass proteins, an overwhelming majority (88%) contain an odd number of TMDs, with most of them being members of the exceptionally successful family of 7-TMD G protein–coupled receptors (GPCRs) (Fig. 1 B). In contrast, $N_{cyt}$ multipass proteins are heavily biased toward even-numbered TMDs (89%).

Inspection of extramembrane domains (i.e., N- and C-terminal tails and inter-TMD loops) showed that the mean exoplasmic (or "outside") domain length was 103 aa (median 17), and a very broad range from zero to over 14,000 aa (Table 1). The distribution of exoplasmic tail lengths among single-pass and multipass proteins was highly asymmetric (Fig. 2 A). Single-pass proteins have a mean exoplasmic tail length of 408 aa (median 277), in contrast to multipass proteins, whose mean exoplasmic tail length is 90 aa (median 24). The exoplasmic loops of multipass proteins are markedly shorter, with a mean of 23 aa (median 11). In total, 76% (1,630 out of 2,137) of long exoplasmic domains (defined as >100 aa) across the membrane proteome are found in single-pass proteins, even though they comprise ~46% of all membrane proteins and contain only 18.5% of all exoplasmic domains.

This length bias was similar but less extreme for cytosolic domains due to longer tails for multipass proteins and shorter tails for single-pass proteins relative to their exoplasmic counterparts (Table 1 and Fig. 2 B). Single-pass proteins contain cytosolic tails with a mean length of 188 aa (median 76). In contrast, the mean cytosolic tail length for multipass proteins is 95 aa (median 41). The mean cytosolic loop length for multipass proteins is substantially shorter at 28 aa (median 15), albeit with a distribution that is longer than that for exoplasmic loops (Fig. 2 C).

Thus, as observed in early genome-wide analyses across multiple organisms (Wallin and von Heijne, 1998), single-pass proteins as a group generally protrude much further away from the membrane surface than multipass proteins, most regions of which remain close to the membrane. Indeed, 83% of multipass proteins have no long outside loops or tails. This explains why single-pass proteins contain 71% of all usable potential N-glycosylation sites (i.e., noncytosolic sequons located at least 12–14 aa away from a TMD [Nilsson and von Heijne, 1993]). On the other side of the membrane, the longer cytosolic tails of single-pass proteins may reflect a general need for membrane-proximal enzymatic activities (e.g., cytochrome p450s), phosphorylation (e.g., receptor tyrosine kinases), cytoskeletal anchoring (e.g., cadherins), and organelle contact sites (e.g., junctophilins). Single-pass proteins are the primary anchor for most of the largest membrane-proximal exoplasmic and cytosolic domains.

## Multipass proteins are mostly built of TMD pairs

Setting aside the over 800 GPCRs for a moment, a striking pattern among multipass proteins is the strong bias toward an $N_{cyt}$ topology (Fig. 1 A), an even number of TMDs (Fig. 1 B), and the preponderance of short inter-TMD loops (Fig. 2 C). The topologic unit that is by far the most common within all multipass proteins (including GPCRs) is the TMD pair: a motif we define as two TMDs separated by a short (≤50 aa) exoplasmic loop. Of the 8,388 examples of two TMDs connected by an exoplasmic loop, 7,700 (92%) are TMD pairs, 441 (5.3%) have a loop between 51 aa and 100 aa, and 247 (2.9%) have loops >100 aa (Fig. 3 A). This accounts for 91.7% of the 18,298 TMDs found in all multipass proteins. The striking rarity of long exoplasmic loops in multipass proteins has been noted across organisms in early genome-wide analyses (Wallin and von Heijne, 1998). Thus, the strong bias toward even-numbered TMDs in non-GPCR multipass proteins is because they are mostly comprised of a series of TMD pairs linked by short cytosolic loops.

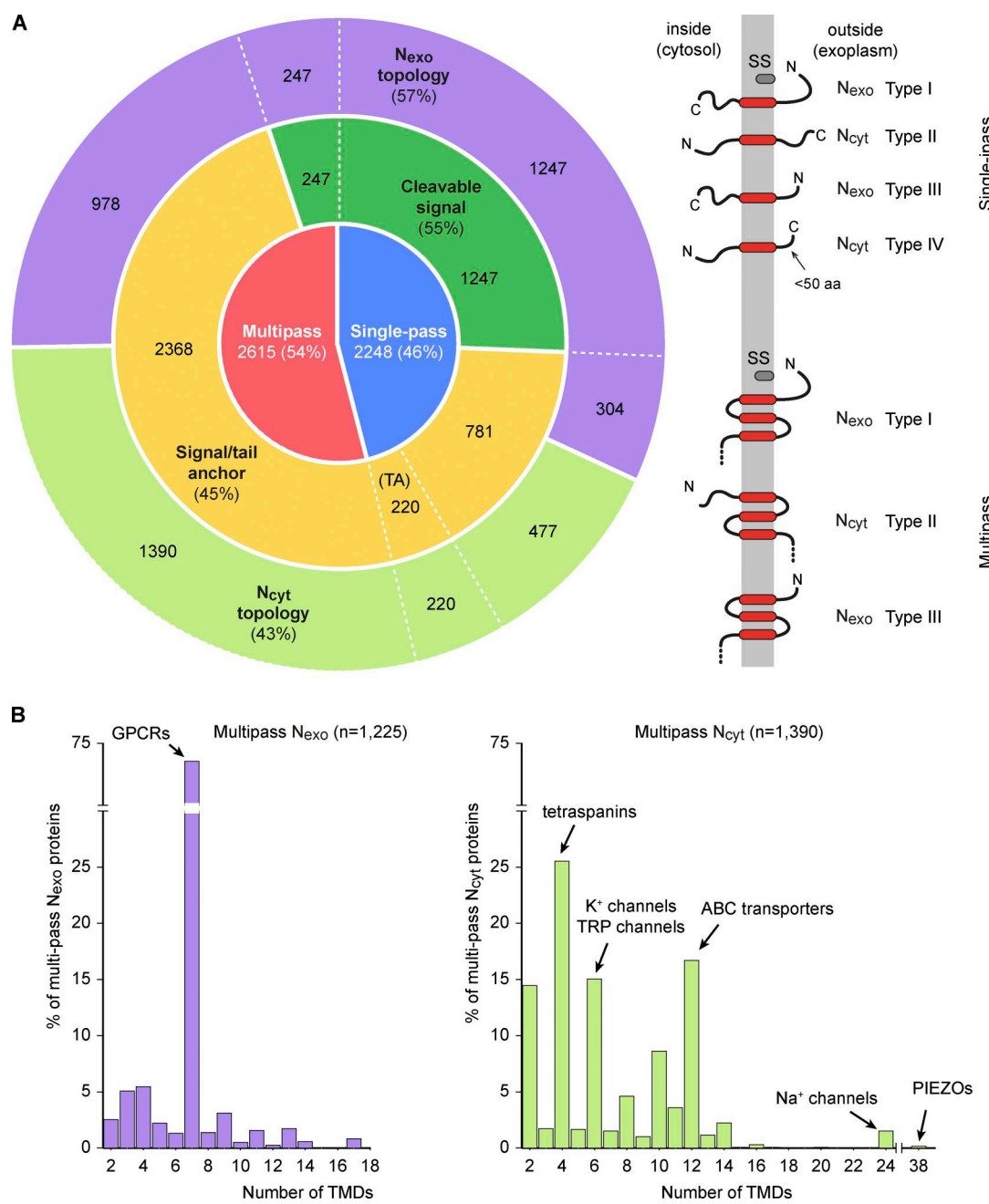

Figure 1. **Topology classification of the human membrane proteome. (A)** Left: classification of the 4,863 human membrane proteins by topology. The innermost ring shows the split between single-pass (blue) and multipass (red) proteins. The middle ring divides each group by targeting mechanism: proteins with a cleavable SS (green) versus those using signal or TA (yellow). The outer ring subdivides proteins by orientation (N terminus exoplasmic, $N_{exo}$, purple; N terminus cytosolic, $N_{cyt}$, green). Right: schematic diagrams of single-pass and multipass proteins that are classified as Type I (cleavable signal, $N_{exo}$), Type II ($N_{cyt}$ SA), Type III ($N_{exo}$ SA), or Type IV (TA, $N_{cyt}$, short C-terminal tail ≤50 aa). **(B)** Distribution of multipass by the number of TMDs. $N_{exo}$ multipass proteins (left graph) are strongly biased toward odd TMD numbers, with 7-TMD proteins dominating. $N_{cyt}$ multipass proteins (right graph) show a strong bias toward even TMD numbers. Notable examples of protein families within each category are indicated.

Topologically, GPCRs can be conceived as three TMD pairs preceded by a single $N_{exo}$ TMD. Although ~11% of GPCRs have large extracellular domains, most do not. Notably however, large extracellular domains are located almost exclusively at the N terminus (typically preceded by a cleavable SS) rather than in a downstream loop (Wallin and von Heijne, 1995). Indeed, of the ~2,500 downstream exoplasmic loops in GPCRs, only one

exceeds 100 aa. Across all multipass proteins, only 1.9% (132 of 6,998) of exoplasmic loops between two downstream TMDs are long (Fig. 3 A). In contrast, 8.3% of exoplasmic loops between TMD1 and TMD2 of $N_{cyt}$ proteins are long, although short loops are still strongly favored. Thus, evolutionary acquisition of long exoplasmic domains in multipass proteins occurs most often before or after TMD1, and rarely between two downstream

Table 1. **Lengths of cytosolic and exoplasmic loops and tails of membrane proteins**

| Location | Topology | Type | Number | Mean (aa) | Median (aa) |
|---|---|---|---|---|---|
| Exoplasmic (outside) | Single-pass | Tails | 2,248 | 408 | 277 |
| | Multipass | Tails | 1,522 | 90 | 24 |
| | | Loops | 8,388 | 23 | 11 |
| | | Both | 9,910 | 34 | 13 |
| | All | Both | 12,158 | 103 | 17 |
| Cytosolic (inside) | Single-pass | Tails | 2,248 | 188 | 76 |
| | Multipass | Tails | 3,708 | 95 | 41 |
| | | Loops | 7,295 | 28 | 15 |
| | | Both | 11,003 | 51 | 19 |
| | All | Both | 13,251 | 74 | 20 |

TMDs. As discussed later, the reason may relate to the very low hydrophobicity of downstream TMD pairs relative to TMD1.

The cytosolic loops connecting TMDs in multipass proteins tend to be short (Fig. 2 C) (Wallin and von Heijne, 1998). Taking the cutoff as 50 aa, 89% (6,520 of 7,295) of cytosolic inter-TMD loops are short, whereas only 4.4% are >100 aa (Fig. 3 B). This is biologically noteworthy because multipass proteins are synthesized at the multipass translocon (MPT), a large ribosome-associated assembly of several membrane-embedded complexes (Smalinskaitė et al., 2022; Sundaram et al., 2022). The MPT provides a semi-protected lipid-filled cavity near the ribosome exit tunnel for TMD insertion, but has relatively little space between the membrane and ribosome to accommodate long cytosolic loops. Indeed, ribosome profiling experiments show that the MPT disassembles during long cytosolic loop synthesis, requiring reengagement of this machinery when additional TMDs are produced (Sundaram et al., 2025). The need for such gymnastics to accommodate long inter-TMD cytosolic loops may explain why they are strongly disfavored in multipass proteins.

TMD pairs seem to insert in a Sec61-independent process, which is speculated to involve the Oxa1 family insertases EMC and GEL (Smalinskaitė et al., 2022; Sundaram et al., 2025), the latter of which is part of the MPT (Hegde and Keenan, 2024). It was speculated long ago that a "helical hairpin" of two closely spaced TMDs could even insert unassisted into the membrane in a concerted manner (Engelman and Steitz, 1981; Steitz et al., 1982). Thus, a sequence of TMD pairs connected by short cytosolic loops, as characterizes most multipass proteins (Fig. 3 B), would insert successively into the lipid cavity of the MPT and never needs to engage the Sec61 lateral gate. In the case of GPCRs, the first TMD would insert via EMC (except the ~11% that have a SS, whose first TMD would insert via Sec61) (Chitwood et al., 2018; Wu and Hegde, 2023), and then, the remaining TMDs would insert as pairs via the MPT. This would explain why most multipass proteins seem to be produced completely normally in the presence of a Sec61 inhibitor, with the only exceptions being the minority with a long translocated tail or loop (Sundaram et al., 2025). As will be shown later, TMD pairs predicted to be inserted via the MPT tolerate exceptionally hydrophilic TMDs beyond what Sec61 is thought to accommodate.

## Composition and properties of TMDs

The 4,863 human membrane proteins contain 20,546 TMDs. The mean TMD length is 23.6 aa (median 24), with 95% ranging from 18 to 30 aa (Fig. 4 A). Notably however, the mean $\Delta G_{app}$ for TMD membrane insertion is +0.59 (median +0.57) (Fig. 4 B), indicating that more than half of all TMDs would be disfavored from Sec61-mediated insertion in isolation (the experimental basis of the $\Delta G_{app}$ scale [Hessa et al., 2005]). The overall aa composition across all TMDs follows the general expectation of being enriched in aliphatic hydrophobic aa, especially Leu, and a lower frequency of polar and charged aa (Fig. 4 C). This distribution of aa, calculated across all human TMDs, differs somewhat from earlier analyses based on more limited datasets derived from membrane protein structures or confidently predicted TMDs (which would bias toward those of higher hydrophobicity) (Arkin and Brunger, 1998; Ulmschneider and Sansom, 2001; Baeza-Delgado et al., 2013).

As expected from earlier observations, both the aa composition (Fig. 4 C) and overall hydrophobicity (Fig. 4, D and E) differ between single-pass and multipass proteins. Single-pass TMDs are highly hydrophobic (mean $\Delta G_{app}$ = –1.89; median –1.82), with 88% of TMDs predicted to favor insertion ($\Delta G_{app}$ < 0). In contrast, multipass TMDs are significantly less hydrophobic (mean $\Delta G_{app}$ = +0.89; median +0.84), with 66% predicted to disfavor insertion ($\Delta G_{app}$ > 0). Consistent with this, hydrophobic aa (Leu, Val, Ile, Ala, Phe, Met, Trp) make up 71% of single-pass TMDs, but only 61% of multipass TMDs. This difference is made up by a higher proportion of polar and charged aa, the latter of which we discuss next.

## Charged aa are prevalent in multipass proteins

Although the most strongly disfavored aa for lipid bilayer insertion are charged (Lys, Arg, Glu, Asp), their appearance in multipass TMDs is not as uncommon as previously thought (Baeza-Delgado et al., 2013; Baker et al., 2017), presumably because many of the most hydrophilic TMDs of multipass proteins were not readily predicted before. TMD charge distribution was visualized by plotting all charged residues in TMDs at the appropriate depth within a schematic membrane. The TMDs of 2,248 single-pass proteins and 2,248 randomly sampled TMDs

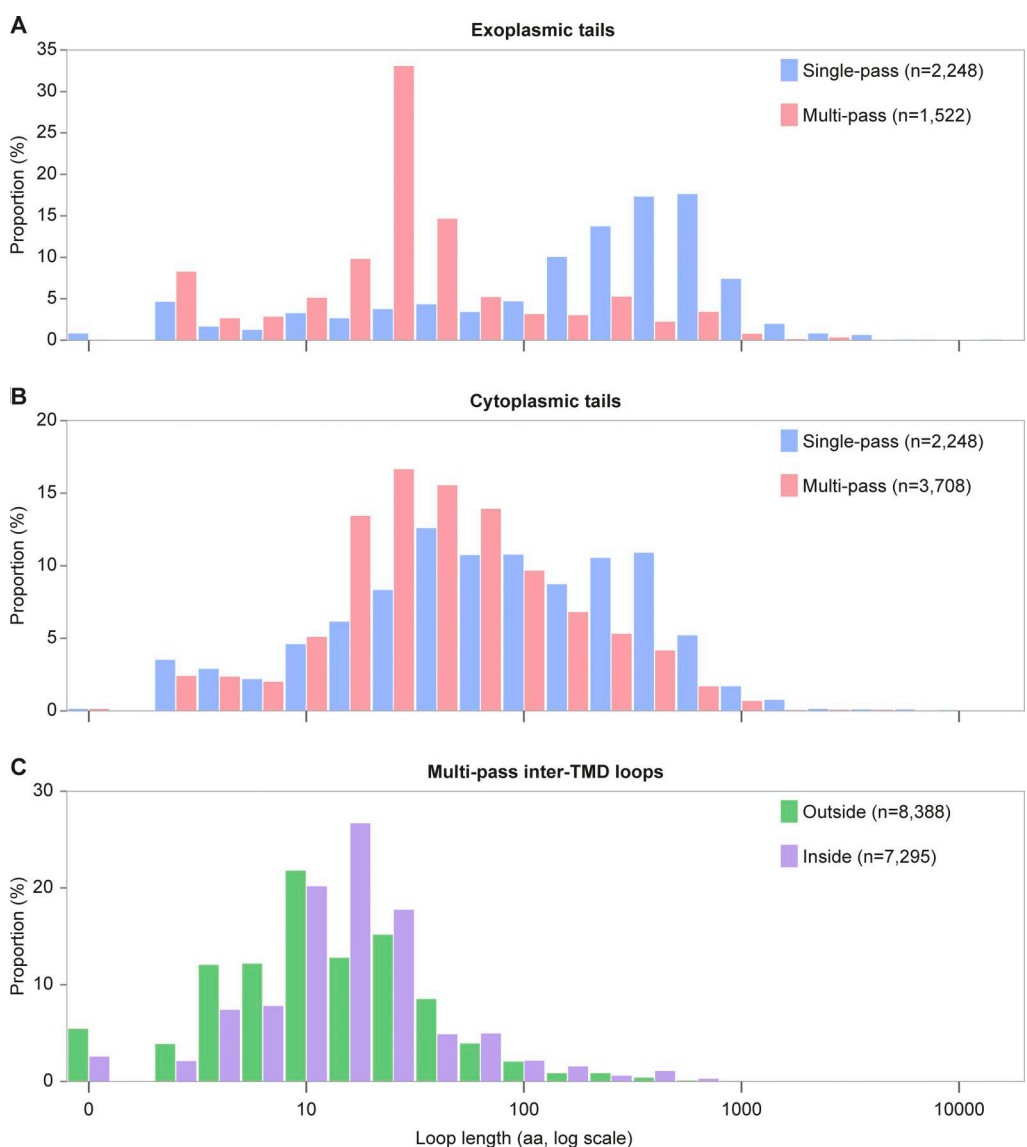

Figure 2. **Length distributions of membrane protein loops and tails. (A–C)** Length distributions of exoplasmic tails (A), cytosolic tails (B), and inter-TMD loops (C) in the human membrane proteome. All panels use a logarithmic x axis displaying loop length in aa. Zero-length loops are included in the first bin.

from multipass proteins (to make visual comparisons easier) were plotted. The central 60% of the membrane was assigned as the hydrophobic core of the membrane, with each flanking 20% designated as the partially hydrophobic interfacial regions (White and Wimley, 1999).

Focusing first on single-pass proteins, we found that ∼12% of TMDs contain at least one charged residue within the hydrophobic core (Fig. 5 A), comprising 1% of all aa in the core region. As expected, the central-most part of the hydrophobic core had the lowest charge density (0.5%). Notably, basic residues are favored almost 3:1 over acidic residues within the hydrophobic core, consistent with acidic residues being the least favored to partition into a hydrophobic environment (Wimley and White, 1996). Charged residues are ∼7.1-fold more frequent in the interfacial regions compared with the core, indicating that they are much more tolerated near the edges of the TMD than in the center. This is presumably due to the more amphipathic

character of this part of the lipid bilayer and the capacity of Arg and Lys to "snorkel" away from the hydrophobic core toward the hydrophilic head groups (Segrest et al., 1990; Strandberg and Killian, 2003).

The frequency and distribution of charged residues in multipass protein TMDs were notably different from single-pass TMDs (Fig. 5 B). First, charged residues are 3.4-fold more likely to be found in the hydrophobic core of multipass TMDs than single-pass TMDs, and were distributed uniformly across the core region. Approximately 36% of TMDs and 86% of all multipass proteins contain at least one charged residue in the core. Second, the 3:1 bias of basic to acidic residues in the hydrophobic core seen in single-pass TMDs is instead biased slightly toward acidic residues in multipass TMDs. The higher frequency and lack of bias are probably because charged residues in multipass proteins typically face the interior of multi-TMD bundles, avoiding exposure to the lipid bilayer. For this

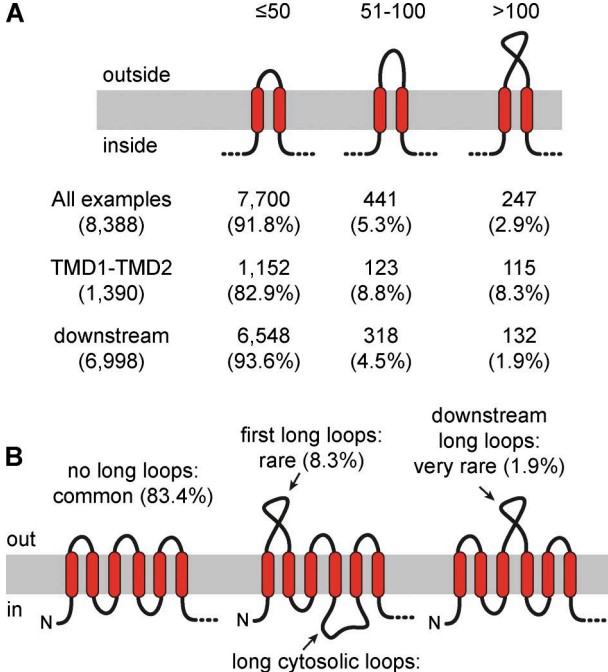

**A**

|  | ≤50 | 51-100 | >100 |
|---|---|---|---|
| All examples (8,388) | 7,700 (91.8%) | 441 (5.3%) | 247 (2.9%) |
| TMD1-TMD2 (1,390) | 1,152 (82.9%) | 123 (8.8%) | 115 (8.3%) |
| downstream (6,998) | 6,548 (93.6%) | 318 (4.5%) | 132 (1.9%) |

**B**

Figure 3. **Prevalence of short and long inter-TMD loops in multipass proteins. (A)** Classification of all exoplasmic inter-TMD loops in multipass proteins by length. Counts and percentages are shown for all outside (exoplasmic) inter-TMD loops, outside loops between TMD1 and TMD2, and outside loops downstream of TMD2. **(B)** Summary of loop length prevalence in multipass proteins. Most multipass proteins do not contain any long outside or inside loops. Percentages of first outside loops, downstream outside loops, and cytosolic loops that exceed 100 aa are indicated.

reason, the highly asymmetric distribution of charged residues between the interfacial versus core regions seen in single-pass TMDs (a ~7.1-fold difference) is less severe in multipass TMDs (~2.5-fold difference).

The frequency of acidic and basic residues differs between the exoplasmic and cytosolic interfacial regions. Whereas the exoplasmic region has roughly equal numbers of both charges, the cytosolic region strongly favors basic residues. This bias for positive charges is particularly strong for single-pass TMDs (89% basic, 11% acidic), but also clear for multipass TMDs (73% basic, 27% acidic). The mean net charge per TMD in the cytosolic interfacial region for single-pass and multipass proteins is +0.30 and +0.17, respectively. The slightly weaker average bias for multipass TMDs may reflect the fact that most of their TMDs are constrained in their topology by adjacent TMD(s), thereby relying less on any single topologic determinant. The net positive charge in the cytosolic interfacial region, together with an even greater positive charge enrichment in the adjacent cytosolic flanking segment (discussed next), reflects the long-observed positive-inside rule.

### The positive-inside rule is weaker for individual TMDs of multipass proteins

The average net charge within 5, 10, or 15 aa on the cytosolic side of single-pass TMDs is +1.30, +2.04, and +2.21. For the exoplasmic side, the average net flanking charge for 5, 10, and 15 aa is

−0.04, −0.17, and −0.26. For the TMDs of multipass proteins, the net charge in the cytosolic flanking segments is +0.53, +0.93, and +1.22, and in the exoplasmic flanking segments is −0.05, −0.12, and −0.20. Thus, as with the interfacial region, the magnitude of the positive-inside bias is stronger for single-pass TMDs relative to multipass TMDs, and is mostly contained within the first 10 flanking residues. A very small bias toward negative-outside charge is seen for both single-pass and multipass proteins (Wallin and von Heijne, 1998). A histogram of charge distributions for single-pass versus multipass proteins combining the interfacial and flanking 10 aa for the inside and outside parts of a TMD illustrates the magnitude of the positive-inside rule (Fig. 5 C).

Importantly, this pattern is also seen for proteins of well-studied families whose topology was unambiguously assigned without consideration of flanking charge. For example, the 166 single-pass SA proteins from three such families (cytochrome P450s, glycosyltransferases, and E3 ligases) showed average net exoplasmic and cytosolic charges of −0.13 and +2.51, respectively. For the SAs from topologically unambiguous multipass proteins (GPCRs, E3 ligases, ABC transporters, channels, solute carriers, and ATPases), the average net exoplasmic charge is −0.12 and net cytosolic charge is +1.16. Thus, our use of positive charge as part of the selection criteria for topology assignment of some proteins has not influenced this analysis of charge distribution.

TMDs have historically been thought to be inserted sequentially, with each TMD adopting the opposite orientation of the preceding one (Blobel, 1980; Wessels and Spiess, 1988). Indeed, TMDs can even be forced into a nonpreferred topology by adjacent TMDs (Ota et al., 1998; Öjemalm et al., 2012; Gafvelin and von Heijne, 1994). Thus, the first TMD was initially thought (and still frequently cited) to be critical for setting overall topology, suggesting that TMD1 of multipass proteins might have a stronger positive-inside bias similar to single-pass TMDs. However, the positive charge bias on the cytosolic side was +1.00 for TMD1 (compared with +2.34 for single-pass proteins) and +1.12 for downstream TMDs of multipass proteins (Fig. 5 D). TMD1 showed a small negative charge bias on the exoplasmic side of −0.38, and this was only −0.06 for downstream TMDs.

These results show that not only is the positive-inside bias very weak for the TMDs of multipass proteins, but only modest differences are seen in flanking charges for the first versus downstream TMDs. This suggested that the parameters that determine topology of multipass proteins may be more distributed than simply TMD1, or are different for different classes of proteins. Consistent with this idea, it was not possible to flip the topology of a multipass protein simply by flipping TMD1 via changes to its flanking charges (Sato et al., 1998). Because multipass proteins have very closely spaced TMDs (Fig. 2 C), the charge biases of adjacent TMDs can potentially sum together to help direct topology. Indeed, much of the apparent discrepancy in a relatively weak positive-inside bias of multipass proteins can be resolved when we consider the properties of TMD pairs. As shown below, the charge properties of the TMD pair considered as a single unit show a positive-inside bias comparable to that seen in the TMDs of single-pass proteins.

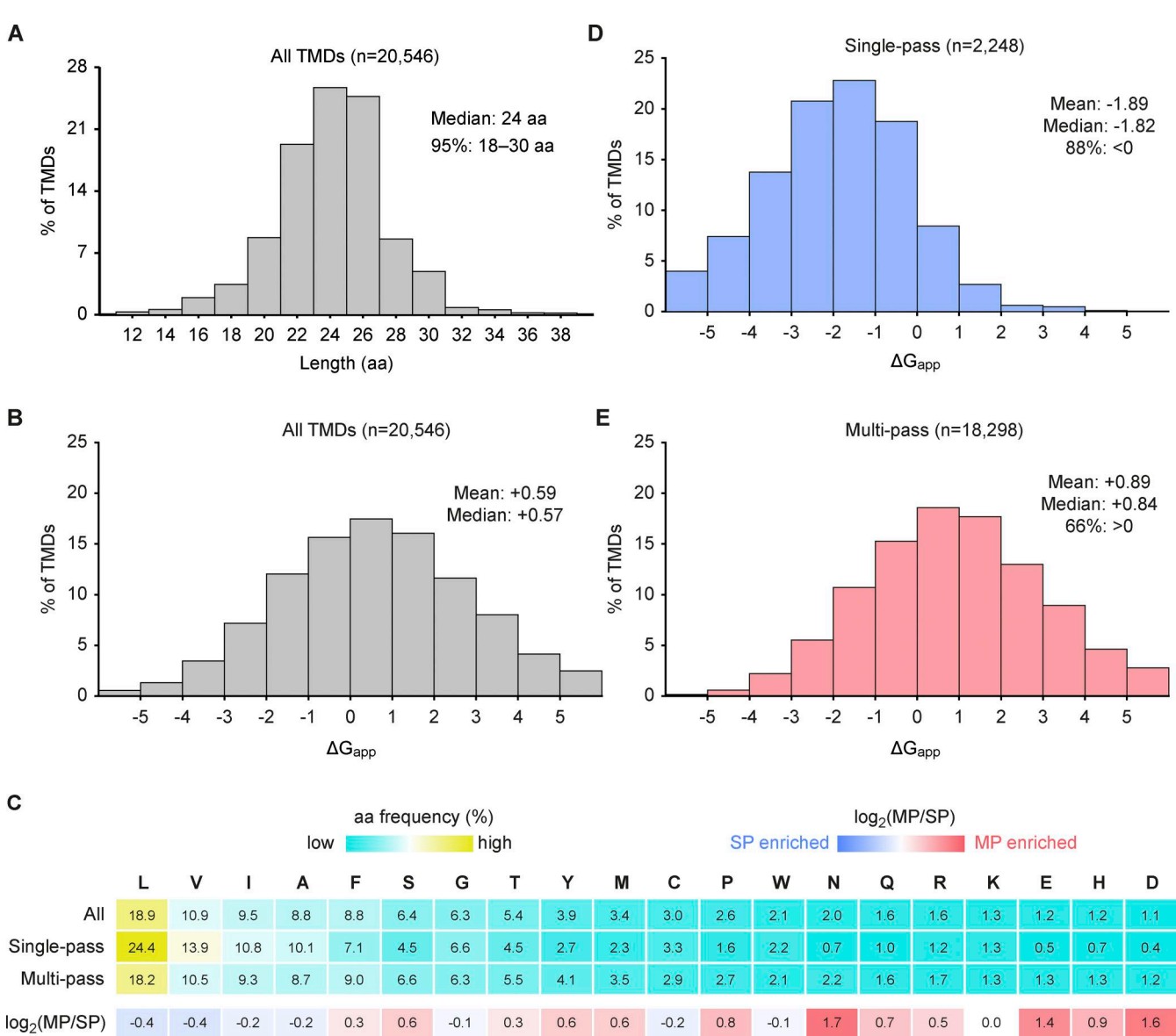

Figure 4. **Biophysical properties of human TMDs. (A)** Histogram of length in aa of all human TMDs. **(B)** Histogram of apparent free energy of membrane insertion ($\Delta G_{app}$, kcal/mol) for all TMDs. **(C)** aa frequency heatmap for all TMDs, subdivided into SP and MP proteins, with aa ordered by decreasing frequency across all TMDs. The top three rows show aa frequency (%) with a color scale from low (cyan) to high (yellow). The bottom row shows $\log_2$(MP/SP) frequency ratios, with blue indicating a higher frequency in SP TMDs and red indicating a higher frequency in MP TMDs. **(D and E)** As in B but for SP (D) and MP (E) proteins. SP, single pass; MP, multipass.

## SA properties reflect their different insertion mechanisms

The position and context of a TMD determine its mechanism of insertion. Two types of insertion machinery, the Sec61 translocation channel or an Oxa1 family insertase, are involved. Sec61 inserts TMDs that precede and follow a long (>100 aa) translocated domain, whereas Oxa1 family insertases generally insert TMDs whose flanking translocated domain is short (<50 aa) (reviewed in Hegde and Keenan, 2024). We used current knowledge of these different insertion routes to segregate human TMDs into different classes and determine their relative lengths, hydrophobicity, and flanking charges. This systematic analysis linking TMD properties to their predicted route of insertion not only supports the idea that their insertion mechanisms differ, but also illuminates the substrate preferences and limits of each mechanism.

We first analyzed the cotranslationally inserted SAs in three categories (Fig. 6 A): $N_{exo}$ (1,282); $N_{cyt}$-long, with a translocated downstream loop >100 aa (558); and $N_{cyt}$-pair, an $N_{cyt}$ SA that is part of a TMD pair (1,152). We found that $N_{exo}$ SAs, most of which are from GPCRs, were notably longer by ~2 aa than either $N_{cyt}$-long or $N_{cyt}$-pair SAs (25 versus 23 aa), consistent with experiments showing that longer hydrophobic segments favor the $N_{exo}$ topology (Sakaguchi et al., 1992; Wahlberg and Spiess, 1997). More strikingly, $N_{cyt}$-pair SAs were markedly less hydrophobic than the other two, with a mean $\Delta G_{app}$ of +0.47. Between $N_{exo}$ and $N_{cyt}$-long SAs, the latter was more hydrophobic (mean $\Delta G_{app}$ of −0.80 and −1.36, respectively). $N_{exo}$ SAs showed both a positive-inside charge bias (+1.29) and a negative-outside bias (−0.54). In contrast, $N_{cyt}$-long SAs showed an even stronger

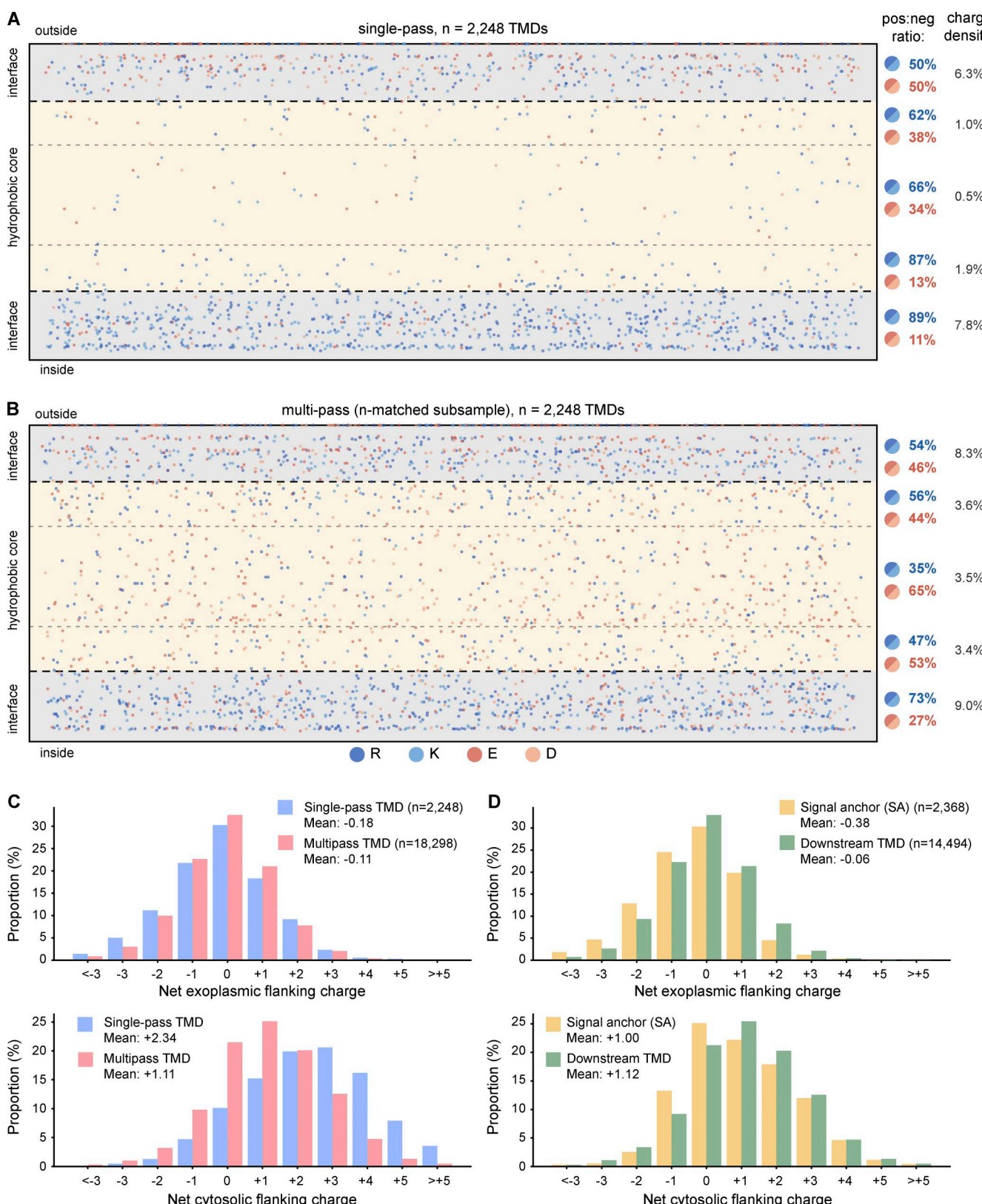

Figure 5. **Distribution of charged residues within and flanking TMDs. (A and B)** Dot plots showing the positions of all charged residues within TMDs of single-pass proteins (A) and an n-matched random subsample of multipass TMDs (B). Each dot represents a single charged residue (R, dark blue; K, light blue; E, dark red; D, light red) plotted at its relative depth within a schematic membrane cross-section. The membrane is divided into an exoplasmic interfacial region (top 20%, light gray shading), hydrophobic core (central 60%, beige shading), and cytosolic interfacial region (bottom 20%, light gray shading), Positive:negative charge ratios and overall charge densities (i.e., percent aa that are charged) are shown for each region on the right. **(C)** Distributions of net exoplasmic (top) and

cytosolic (bottom) flanking charges (the sum of charged residues in the interfacial region of the TMD plus the 10 flanking residues on each side) for single-pass (blue) and multipass (red) TMDs. **(D)** As in C, but for SA TMDs (first TMD of proteins lacking a cleavable signal; yellow) and downstream TMDs (all subsequent TMDs in multipass proteins; green).

positive-inside bias (+2.08), but no negative-outside bias. $N_{cyt}$-pair SAs showed positive-inside and negative-outside biases (+0.89 and –0.28), but less so than $N_{exo}$ SAs. However, when the $N_{cyt}$-pair SA is considered together with its partner TMD, the positive-inside and negative-outside biases are stronger: +1.90 and –0.53, respectively.

The different properties observed among the three types of SAs can be rationalized by their different insertion mechanisms at the ER. After release from SRP, $N_{exo}$ SAs and TMD pairs can be inserted by EMC, whereas $N_{cyt}$-long SAs must be rejected by EMC and inserted by Sec61 (Wu and Hegde, 2023). EMC's cytosolic vestibule, where a TMD's flanking domain resides before translocation, is positively charged. Thus, the particularly strong positive charge bias preceding the hydrophobic core of $N_{cyt}$-long SAs would disfavor EMC engagement (Pleiner et al., 2023), thereby avoiding mis-insertion and allowing triage to Sec61. Conversely, EMC engagement would be favored by the modest negative charge bias flanking the hydrophobic cores of $N_{exo}$ and $N_{cyt}$-pair SAs, whereas mis-insertion would be disfavored by the positive charge bias in the other flanking segment.

The low hydrophobicity of $N_{cyt}$-pair SAs can be explained by a model in which they can insert in concert with a nearby downstream TMD, thereby shielding some of their hydrophilic regions while being stabilized in the membrane by their combined hydrophobicity. In contrast, the high hydrophobicity of $N_{exo}$ and $N_{cyt}$-long SAs is because they need to enter the membrane in isolation before any potential partner TMDs have been synthesized. Notably, when $N_{exo}$ and $N_{cyt}$-long SAs were each segregated by single-pass versus multipass, single-pass proteins showed a substantially greater positive-inside bias and lower $\Delta G_{app}$ (Fig. 6 B). This can be rationalized by the fact that single-pass proteins rely exclusively on this TMD and its flanking regions for topology determination, and, in most cases, also need to reside in the membrane stably without subsequent TMD-TMD interactions.

### A preceding TMD facilitates insertion of the next TMD through Sec61

An N-terminal SS or an $N_{cyt}$-long SA engages Sec61 and initiates translocation through its central channel. Thus, the TMD that follows a SS or $N_{cyt}$-long SA necessarily accesses the membrane via Sec61's lateral gate (Fig. 7 A). Analysis of these obligately Sec61-dependent TMDs showed that their properties were starkly different depending on whether they were preceded by a cleavable SS or $N_{cyt}$-long SA. Those preceded by a SS have a high average hydrophobicity ($\Delta G_{app}$ of –1.89) with a strong positive-inside bias (+2.41) and a weak negative-outside bias (–0.27). Only 9.4% of these TMDs have a positive $\Delta G_{app}$ value. In contrast, TMDs that follow an $N_{cyt}$-long SA have an average $\Delta G_{app}$ value of 0, with just over half having a positive value. Furthermore, the positive-inside bias was weaker (+1.17), with no negative-outside bias (+0.09).

These findings indicate that insertion of a TMD through the Sec61 lateral gate in the absence of any preceding TMDs in the membrane follows the expectations for passive membrane partitioning based on hydrophobicity. Indeed, single-pass TMDs in this class show the highest hydrophobicity and greatest flanking charge bias of all TMD classes analyzed (Fig. 7 B). The strong positive charge bias downstream of these TMDs may serve to pause translocation just as the TMD enters the Sec61 channel, thereby providing more time for the TMD to pass through the lateral gate into the lipid bilayer. Consistent with this model, tandem positively charged residues slow polypeptide passage through Sec61 (Yamagishi et al., 2014).

Although the first TMD of multipass proteins preceded by a SS is also hydrophobic on average ($\Delta G_{app}$ of –0.80), it is less so than single-pass proteins (Fig. 7 B). Presumably, these TMDs can afford to be temporarily less stable in the membrane because later TMDs in the protein can stabilize them via TMD-TMD interactions. The markedly higher average $\Delta G_{app}$ (and weaker positive-inside bias) of a TMD preceded by a $N_{cyt}$-long SA supports the idea that an already-inserted TMD can facilitate insertion of the next one. TMD cooperation during insertion by Sec61 has been suggested previously using model proteins (Heinrich and Rapoport, 2003).

### High TMD hydrophilicity is tolerated in TMD pairs of multipass proteins

To further examine the idea of TMD-TMD cooperation during insertion, we analyzed the highly prevalent TMD-pair motif, with 7,700 instances in the membrane proteome. Recent findings (Smalinskaitė et al., 2022; Sundaram et al., 2025) have suggested a model (Hegde and Keenan, 2024) in which the TMD pair may insert concertedly as a unit, or at least with each TMD inserting in rapid succession. Indeed, two closely spaced TMDs have been observed to interact with each other near the ribosome surface prior to insertion (Tu et al., 2014). A concerted insertion model suggests that their TMDs would, as a general class, tolerate hydrophilic character more than any other types of TMDs. We therefore analyzed the properties of all TMD pairs, as well as three subclasses based on their topologic context (Fig. 8). Due to the close TMD juxtaposition (mean loop length 13.8 aa, median 10 aa; Fig. 2 C), the inside charge bias was summed for the two TMDs, and the outside charge bias was calculated from the entire exoplasmic loop plus exoplasmic interfacial regions of both TMDs.

The mean $\Delta G_{app}$ for the TMDs in the 7,700 TMD pairs was +1.12, with the second of the pair being slightly more hydrophilic than the first. Subclassification of TMD pairs by the first pair versus downstream pairs shows that TMDs forming the TMD1-TMD2 pair are less hydrophilic than the TMDs of internal pairs following a short cytosolic loop. This difference presumably reflects the fact that the TMD1-TMD2 pair needs to mediate targeting (particularly TMD1, which is the more hydrophobic of

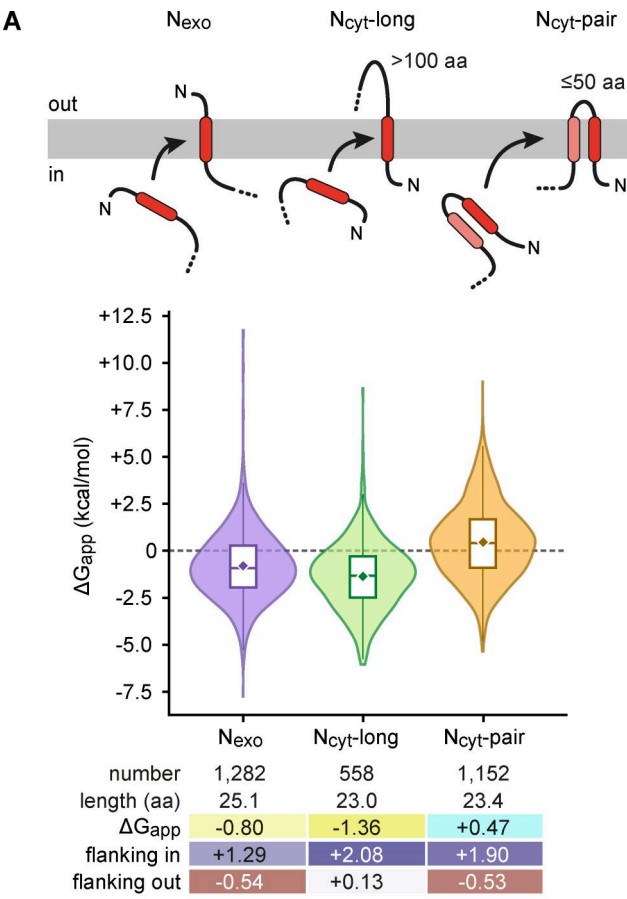

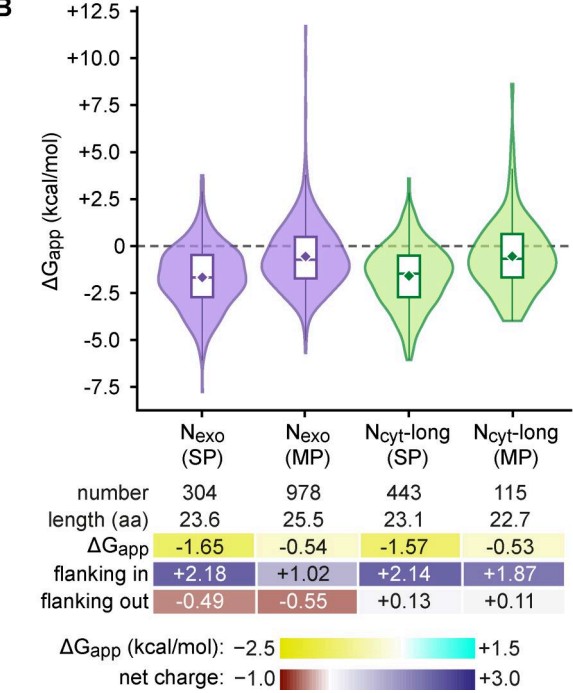

| | $N_{exo}$ | $N_{cyt}$-long | $N_{cyt}$-pair |
|---|---|---|---|
| number | 1,282 | 558 | 1,152 |
| length (aa) | 25.1 | 23.0 | 23.4 |
| $\Delta G_{app}$ | -0.80 | -1.36 | +0.47 |
| flanking in | +1.29 | +2.08 | +1.90 |
| flanking out | -0.54 | +0.13 | -0.53 |

| | $N_{exo}$ (SP) | $N_{exo}$ (MP) | $N_{cyt}$-long (SP) | $N_{cyt}$-long (MP) |
|---|---|---|---|---|
| number | 304 | 978 | 443 | 115 |
| length (aa) | 23.6 | 25.5 | 23.1 | 22.7 |
| $\Delta G_{app}$ | -1.65 | -0.54 | -1.57 | -0.53 |
| flanking in | +2.18 | +1.02 | +2.14 | +1.87 |
| flanking out | -0.49 | -0.55 | +0.13 | +0.11 |

$\Delta G_{app}$ (kcal/mol): −2.5 to +1.5
net charge: −1.0 to +3.0

Figure 6. **Properties of SA TMDs. (A)** Comparison of the properties of three classes of cotranslationally inserted SA TMDs: $N_{exo}$ (purple), $N_{cyt}$-long (green), and $N_{cyt}$-pair (yellow). Violin plots are shown below the respective diagrams. In this and subsequent figures, the violin plots show $\Delta G_{app}$ distributions for each class, with embedded box plots indicating mean (diamond), median, and interquartile range. Summary statistics are shown below the violin plots, colored according to the heatmap scales for hydrophobicity and net charge under B. **(B)** As in A, but subdivided into SP and MP groups. SP, single pass; MP, multipass.

the two), is the first to insert, and uses an insertase (probably EMC in most cases) in isolation. In contrast, later TMD pairs emerge from a membrane-docked ribosome and insert into the specialized environment generated by the MPT (Sundaram et al., 2022; Sundaram et al., 2025; Smalinskaitė et al., 2022). Not only is the MPT cavity relatively protected to allow TMD-TMD interactions, but is notably hydrophilic on the surfaces of the PAT chaperone complex and GEL insertase (Smalinskaitė et al., 2022).

Ribosome profiling experiments have shown that the MPT disassembles during the translation of long (>100 aa) cytosolic loops (Sundaram et al., 2025), requiring retargeting to the translocation machinery when the next TMD(s) emerge. Notably, the first TMD of a TMD pair that emerges after a long cytosolic loop has a mean $\Delta G_{app}$ of +0.65. This is similar to the hydrophobicity of the first TMD of the TMD1-TMD2 pair, and more hydrophobic than the first TMD of a TMD pair that emerges after a short inside loop. This finding supports the idea that a TMD that follows a long cytosolic loop must retarget to the translocation machinery in a similar way as TMD1 of a TMD1-2 pair initially engages this machinery. Another shared property of the TMD1-2 pair and a TMD pair following a long cytosolic loop is a negative-outside charge bias. This is similar to what is seen with $N_{exo}$ SAs, which is noteworthy because all three elements must (re)initiate insertion (likely via EMC) after (re)targeting.

The strikingly high hydrophilicity of TMD pairs is not simply due to most of them being internal TMDs of multipass proteins. This is illustrated by considering the properties of internal TMDs that precede or follow a long (>100 aa) exoplasmic loop (Fig. 8). Analysis of the 132 examples in multipass proteins shows that both TMDs that flank a long exoplasmic internal loop are markedly more hydrophobic ($\Delta G_{app}$ of −0.92 and −0.49) than TMDs that flank a short internal exoplasmic loop ($\Delta G_{app}$ of +1.18 and +1.24). Ribosome profiling and biochemical experiments show that long exoplasmic loops between internal TMDs trigger disengagement of the MPT and likely engagement of Sec61, through which the loop is translocated (Sundaram et al., 2025). Thus, as seen earlier with the analysis of SAs, the properties of a TMD are strongly influenced by its context and, in turn, by the translocation machinery it engages for its membrane insertion.

## Discussion

Our systematic analysis of the human membrane proteome shows that the properties of a TMD and its flanking region are correlated with its topologic context. We have made various context-dependent TMD comparisons: occurrence in single-pass versus multipass proteins, occurrence at the beginning or middle of a multipass protein, length of flanking regions, and first versus second TMD of a pair. These comparisons revealed

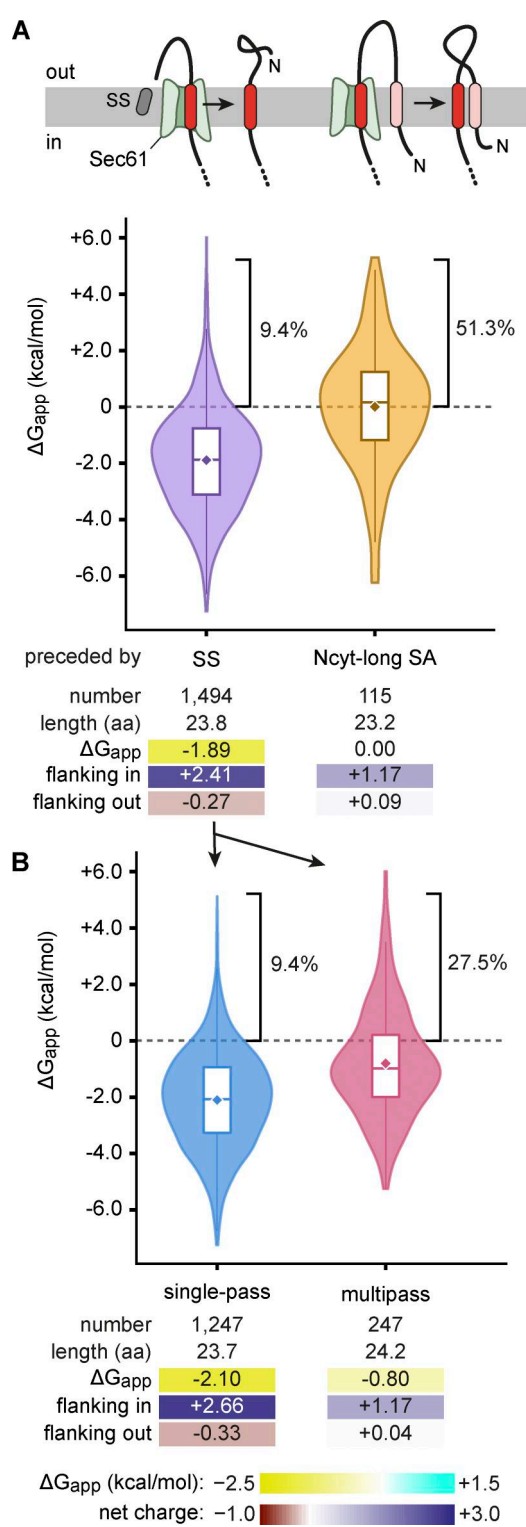

Figure 7. **Properties of TMDs that insert through Sec61. (A)** Properties of TMDs that are downstream of either a cleavable SS or an $N_{cyt}$-long SA. Violin plots and summary statistics are shown below the diagrams. **(B)** As in A, but subdivided into single-pass and multipass proteins.

several key differences among classes of TMDs that can be rationalized with current knowledge of membrane protein insertion pathways. Notwithstanding these nuances, a broader view

suggests that six core membrane insertion reactions can explain biogenesis of the membrane proteome (Fig. 9 A). Four of these reactions apply to both single-pass and multipass proteins, whereas two are uniquely for multipass proteins. Such a synthesis reveals some overarching themes.

Single-pass proteins are inserted by one of four routes, each of which is typically associated with a high-hydrophobicity TMD (average $\Delta G_{app}$ values less than –1.5). Reaction 1 is entirely Sec61-dependent: a SS initiates translocation through Sec61, and a downstream TMD passes through Sec61's lateral gate into the membrane. Reaction 2 is insertion of a $N_{cyt}$ SA via Sec61, through which a long downstream loop is translocated. Reaction 3 is insertion of a $N_{exo}$ SA, typically via EMC. Reaction 4 is the posttranslational insertion of a TA protein using either the GET or EMC insertases for TMDs of high or low hydrophobicity, respectively.

These same four insertion reactions in multipass proteins are associated with TMDs of substantially lower hydrophobicity (although still with negative $\Delta G_{app}$ values for the first three). This systematic difference suggests that these four core single-TMD targeting and insertion reactions have a moderate hydrophobicity requirement, regardless of whether they use Sec61 or EMC. It is likely that the additional hydrophobicity seen in single-pass proteins is due to most of them residing in the membrane in isolation, without the benefit of stabilizing TMD-TMD interactions. Indeed, the most hydrophilic class among single-TMD insertion reactions was the final C-terminal TMD of multipass proteins, which can be stabilized immediately after insertion by all previous TMDs (Wu et al., 2024).

Reaction 5, which is a variation of reactions 1 and 2, is specific to multipass proteins and results in the insertion of two TMDs separated by a long exoplasmic loop. This is obligatorily mediated by Sec61 regardless of where within a multipass protein this two-TMD unit occurs. The TMDs have moderate average hydrophobicity, with the first TMD typically being more hydrophobic than the second. The sixth reaction, insertion of a TMD pair, is overwhelmingly the most common across the entire membrane proteome. In stark contrast to the other five reactions, TMD-pair insertion tolerates remarkably high hydrophilicity in either or both TMDs of the pair. Tolerance for high hydrophilicity in TMD pairs explains why they dominate multipass proteins, whose TMDs have long been known to be highly hydrophilic.

What emerges is a picture in which hydrophilicity-tolerating insertion reactions by the Oxa1 family of insertases, primarily EMC and GEL, build the vast majority of multipass proteins (Fig. 9 B). Only a minority of insertion reactions during multipass protein biogenesis are predicted to obligately rely on Sec61. Conversely, most single-pass proteins rely on Sec61-mediated reactions. This Sec-Oxa dichotomy between single-pass and multipass proteins matches the observation that large exoplasmic domains are found primarily in single-pass proteins, whereas most multipass proteins have only short exoplasmic loops and tails. Thus, the underlying logic is simple: a channel is needed for long-domain translocation, whereas a channel-independent facilitated insertion reaction suffices for short-domain translocation.

Figure 8. **Properties of TMDs separated by a short or long exoplasmic loop.** Cartoons showing the four classes of adjacent TMDs that were analyzed, with the inserting two TMDs in red and already-inserted TMDs in pink. Violin plots for all TMD pairs and for each diagrammed class are shown along with summary statistics.

The positive-inside rule is universal, is seen in the interfacial region of a TMD and ~10 flanking residues, and is typically weaker for individual TMDs of multipass proteins. However, when the fact that most multipass TMDs are part of TMD pairs is taken into account, a similar approximately +2 positive charge bias is seen across the board. In contrast, a modest but clear negative-outside bias of approximately −0.5 is evident primarily for some classes of TMDs inserted by the Oxa1 family: $N_{exo}$ SAs, TMD1-2 pairs, and TMD pairs following a long cytosolic loop. The highly conserved positive charge inside the hydrophilic vestibule of Oxa1 family members likely explains this negative-outside bias, which was previously difficult to discern when all TMDs were analyzed together.

The starkly hydrophilic nature of many TMDs in TMD pairs means insertion of lengthy exoplasmic domains would not be tolerated in most cases. This is because a former TMD pair that now has a long exoplasmic domain would need to switch from an EMC or MPT mechanism to Sec61, a reaction that requires higher hydrophobicity due to sequential TMD insertion. These differential requirements may explain the rarity of long exoplasmic loops, the evolutionary acquisition of which would need multiple simultaneous changes to one or both flanking TMDs to

switch insertion pathways. Knowledge of TMD requirements relative to flanking domains should help engineer membrane proteins in a more rational manner while retaining their compatibility with biogenesis. Our analyses provide a holistic view of the composition, properties, and evolution of the human membrane proteome. Given the high conservation of the Sec61 and Oxa1 family going back to the last universal common ancestor (Lewis and Hegde, 2021), the principles highlighted in our analysis are likely to apply broadly across life.

## Materials and methods

### Protein dataset curation

All human proteins annotated as containing one or more alpha-helical TMDs were extracted from two primary sources: the AFTM database (accessed 2023 [Pei and Cong, 2023]), which derives TMD predictions from AlphaFold2 structural models for alpha-helical membrane proteins, and UniProt (release 2025_02 [UniProt Consortium, 2025]), which integrates curated information on proteins. Both sources combined yielded 5,548 candidate membrane proteins that were the starting point of our

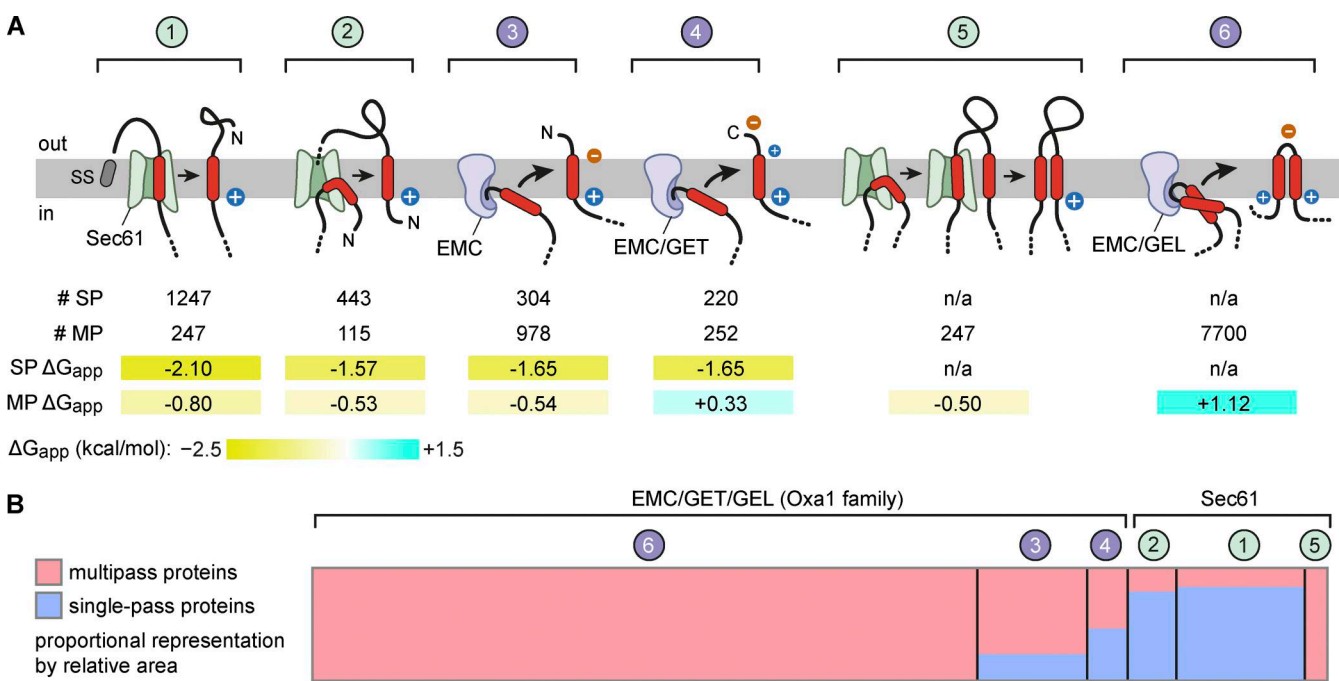

**Figure 9.** **Six core reactions for membrane protein insertion. (A)** Models of the six core insertion reactions that occur during membrane protein biogenesis. Reactions 1–4 apply to both SP and MP proteins; reactions 5–6 only apply to MP proteins. Reactions 1, 2, and 5 require Sec61, whereas reactions 3, 4, and 6 involve Oxa1 family insertases. Charge biases are indicated on the substrate. The number of instances in the human proteome of each reaction and average $\Delta G_{app}$ for those set of TMDs are indicated. **(B)** Proportional area chart showing the relative contribution of each reaction to the total membrane proteome, subdivided by SP (blue) and MP (red) proteins. Reactions are grouped by the two major translocation machineries: EMC/GET/GEL (Oxa1 family, left) versus Sec61 (right). This chart visually illustrates that the majority of Oxa1 family reactions operate on MP proteins, whereas the majority of Sec61-mediated reactions operate on SP membrane proteins. TMD-pair insertion is by far the most common insertion reaction during biogenesis of the membrane proteome. SP, single pass; MP, multipass.

curation process. From the initial list, mitochondrial membrane proteins were removed, as we primarily focus on ER-inserted membrane proteins. Mitochondrial-inserted proteins were previously well curated through the combination of UniProt and MitoCarta 3.0, and cross-reference with both databases led to the removal of 324 mitochondrial proteins, leaving 5,224 candidates.

For each of the 5,224 remaining proteins, the predicted number of TMDs was compared across up to six sources, where available: AFTM (Pei and Cong, 2023), UniProt (UniProt Consortium, 2025), HTPTM (Dobson et al., 2015), TmAlphaFold (Dobson et al., 2023), PDBTM (Kozma et al., 2013), and Membranome (Lomize et al., 2017). A protein was classified as concordant if all available sources agreed on the same number of TMDs ($n$ = 3,307). A protein was classified as conflicted if any discrepancy existed between sources ($n$ = 1,917). Of the 3,307 concordant proteins, 6 were found to correspond to obsolete UniProt entries and were removed. A further 7 were removed because they were annotated as cytosolic or mitochondrial in at least one independent source, had no obvious hydrophobic segment, or had no predicted hydrophobic region in AlphaFold-predicted structures.

Each of the 1,917 conflicted proteins was manually evaluated, and 348 proteins were removed using the following criteria. First, 3 obsolete entries were removed. Second, 45 secreted proteins were removed. Secreted proteins were identified as containing cleavable SS (UniProt annotation) or SignalP-6.0 (Teufel et al., 2022) prediction, having no predicted TMDs by

AlphaFold (helical segment ≥15 aa with a predominantly hydrophobic solvent–exposed surface), and having no predicted TMD by a $\Delta G_{app}$ scan (<0 kcal/mol, for obligate Sec61 lateral gate–based insertion [Hessa et al., 2005]), and UniProt subcellular localization was "secreted" or similar. Third, 290 cytosolic proteins were removed. Cytosolic proteins were identified as candidates lacking cleavable SS, and lacking predicted TMDs as before, except $\Delta G_{app}$ scan cutoff was adjusted to 2.0. Fourth, 10 hairpin proteins were removed. Hairpin proteins were classified as proteins that have hydrophobic segments (<15aa) but do not traverse the membrane fully. Removal of these 348 proteins from the conflicted set left 1,569 proteins, which were added to the 3,294 proteins from the concordant set to yield the final dataset of 4,863 human ER-inserted membrane proteins.

**TMD and SS coordinate assignment**
For each of the 4,863 proteins, TMD start and end coordinates were assigned using the following workflow. First, experimentally determined structures were taken into account to correct inaccurate AFTM TMD coordinates (Kozma et al., 2013). Second, all paralogs were manually examined when TMD number or coordinates are not concordant, in which case all TMDs were manually counted and TMD boundaries are determined with PPM 3.0–based structural models, either available on AFTM or generated on PPM 3.0 server (Lomize et al., 2012). TMDs with unusual lengths (≤14 aa or ≥35 aa) were individually inspected

for accuracy. These TMDs were only retained if they have been reported experimentally in previous literature. Cleavable N-terminal SS were recorded where applicable, starting from position 1 to the position before cleavage site. SS coordinates were mostly derived from UniProt annotation. All proteins that contained long translocated N-terminal tails (>200 aa) were manually inspected for the presence of a SS through SignalP-6.0 (Teufel et al., 2022).

## Topology curation

Each protein was manually inspected, and their topology was determined by the following criteria. First, All GPCRs were assigned $N_{exo}$ topology. Second, all proteins containing cleavable SS were assigned $N_{exo}$ topology. Third, targeted, not large-scale, experimental evidence, such as glycosylation (ER lumen–specific), phosphorylation (cytosolic), protease protection, and experimentally determined structures, was combined to infer topology. Fourth, surface-exposed charges in AlphaFold-predicted models were manually inspected and the side that has more overall positive charges immediately adjacent to TMDs was assigned as the cytosolic side according to the positive-inside rule (von Heijne, 1986; von Heijne, 1989; Hartmann et al., 1989). Lastly, topology assignments were cross-checked with paralogs. Proteins assigned a different topology from all other members of their family were reinspected and reassigned only if strong independent evidence supported the discrepancy.

## Loop and tail coordinate calculation

Coordinates for the flanking loops and tails were deduced from TMD coordinates. For example, for a protein with n TMDs: N-terminal tail (loop 1 in Table S1) is residues 1 to (TMD1_start – 1), or (signal_end + 1) to (TMD1_start – 1) if a signal peptide is present. Internal loops are residues (TMD1_end + 1) to (TMD2_start – 1); (TMD2_end + 1) to (TMD3_start - 1), until the loop preceding TMDn. Due to the overlapping TMDs, some loops will be zero length. C-terminal tail (loop n+1 in Table S1) is residues (TMDn_end + 1) to the end of the protein. Loop location (inside or outside) was also assigned. $N_{cyt}$ topology has N-terminal tail inside, while $N_{exo}$ topology has N-terminal tail outside. The subsequent loops and C-terminal tail follow alternating locations.

## Dataset structure and querying

The complete curated dataset was organized as a single spreadsheet (Table S1) with one row per protein containing all topologic information. Columns are organized into seven categories: (1) protein identity (UniProt accession, gene name, synonyms, length, TMD coordinate string); (2) topology (number of TMDs, topology type, N and C terminus locations); (3) notes for functional classifications (see below); (4) full sequence; (5) signal peptide coordinates and sequences; (6) per TMD coordinates and hydrophilicities (measured by $\Delta G_{app}$); and (7) per loop coordinates, lengths, and locations.

All quantitative analyses were performed by querying this table either directly in Excel or programmatically with Python scripts using the pandas library. Subsets of proteins or TMDs were defined by applying logical filters to the relevant columns (e.g., Number_of_TMDs == 1 for single-pass proteins).

## Functional classifications

Proteins in topology viewer were classified into categories as follows: common essential genes from DepMap (depmap.org); E3 ubiquitin ligases from https://esbl.nhlbi.nih.gov/Databases/KSBP2/Targets/Lists/E3-ligases/; ion and water channels from Taujale et al. (2025); solute carriers from https://slc.bioparadigms.org/; ABC transporters from Vasiliou et al. (2009); and ATPases from UniProt annotations.

## Biophysical property calculations

Apparent free energies of membrane insertion ($\Delta G_{app}$, kcal/mol) were calculated for all TMDs using the $\Delta G$ prediction server (https://dgpred.cbr.su.se/ [Hessa et al., 2005]). For TMDs >40 aa, full protein scans were used to identify the minimal $\Delta G_{app}$ area. All 20,546 TMDs have $\Delta G_{app}$ values recorded in the dataset.

Net flanking charges were calculated by counting basic (K, R) and acidic (D, E) residues within defined windows flanking each TMD boundary. Three windows were used: 5, 10, and 15 residues on each side of each TMD. Proportional charge counts were used for loops or tails shorter than the defined cutoff. For the combined interfacial and flanking charge analysis (used in Figs. 5, 6, 7, and 8), the flanking charge was defined as the sum of net charges in the 10 residues immediately flanking the TMD plus the charged residues in the interfacial region of the TMD itself (the outermost 20% of TMD residues on each side).

For the dot plot visualization of charged residue positions within TMDs, each TMD was divided into five zones: exoplasmic (outside) interface (outermost 20%), three segments of the hydrophobic core (each 15%, 30%, and 15% of TMD length, respectively), and cytosolic (inside) interface (innermost 20%). The position of each charged residue (K, R, D, E) within its TMD was recorded as a fractional positioning. For the n-matched multipass comparison in Fig. 5 B, a random subsample of 2,248 TMDs was drawn.

aa frequencies were calculated as the percentage of each aa type across all residues within the relevant TMD set (all TMDs, single-pass TMDs, or multipass TMDs).

## Statistical analysis

All values reported as means or medians are calculated across all members of indicated categories. No statistical hypothesis tests were applied, as the analyses are reporting qualitative differences and based on near-complete proteome-scale data rather than samples drawn from a larger population.

## Data visualization

Draft figures were generated in Excel or Python using matplotlib. Violin plots were generated with embedded box plots and show the mean (diamond marker), median (horizontal line), and interquartile range. Final figures were compiled in Adobe Illustrator.

## Use of AI assistance

Computational analysis scripts, draft figure generation scripts, draft figure generation, and draft website development involved assistance from Claude (Anthropic). The AI agent was used interactively to write and verify Python analysis code, and

cross-check numerical results against the dataset. All numerical results reported in the paper were independently verified by direct querying of the curated dataset (Table S1). All analyses and scripts were used only with contained, curated dataset described in this manuscript. All TMD analysis, topology assignments, and all scientific interpretations are the work of the authors.

## Online supplemental material
Fig. S1 shows the curation process of the dataset in this article. Fig. S2 shows an overview of the topology viewer and key features. Table S1 contains the topologic parameters for all membrane proteins analyzed in this study.

## Data availability
All data compiled in this manuscript are available from Table S1 and membrane protein topology viewer (topology.bioch.ox.ac.uk).

## Acknowledgments
We thank C. Desroches Altamirano, R. Judy, H. Wang, and Z. Y. Gan for thoughtful discussions and comments on this manuscript.

This work was supported by the UK Medical Research Council (grant MC_UP_A022_1007 to R.S. Hegde) and the Wellcome Trust (Career Development Award 321904/Z/24/Z to H. Wu). Open Access funding provided by University of Oxford.

Author contributions: Haoxi Wu: conceptualization, data curation, formal analysis, funding acquisition, investigation, methodology, project administration, resources, software, supervision, validation, visualization, and writing—original draft, review, and editing. Ramanujan S. Hegde: conceptualization, funding acquisition, investigation, project administration, supervision, validation, visualization, and writing—review and editing.

Disclosures: The authors declare no competing interests exist.

Submitted: 9 April 2026

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

# Supplemental material

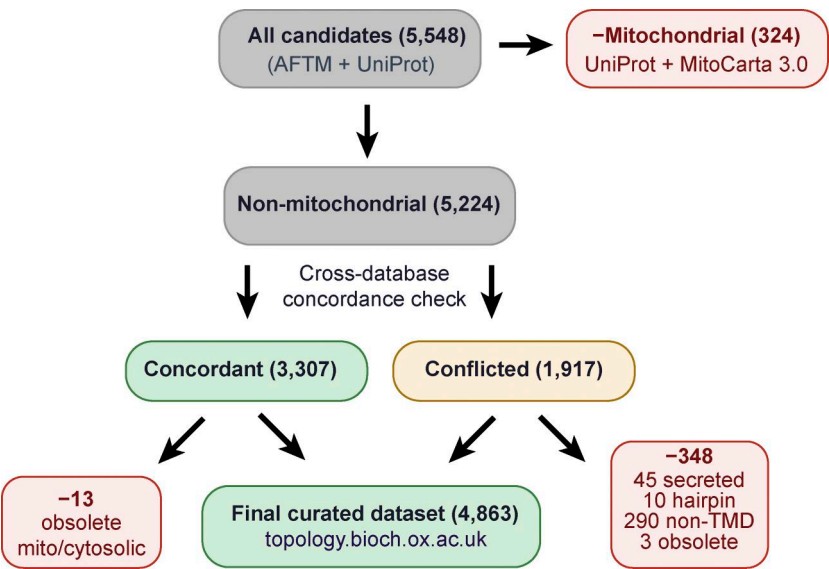

Figure S1. **Curation of the human membrane proteome.** Flowchart summarizing the curation process used to assemble the dataset of human ER-inserted membrane proteins. See text and methods for details.

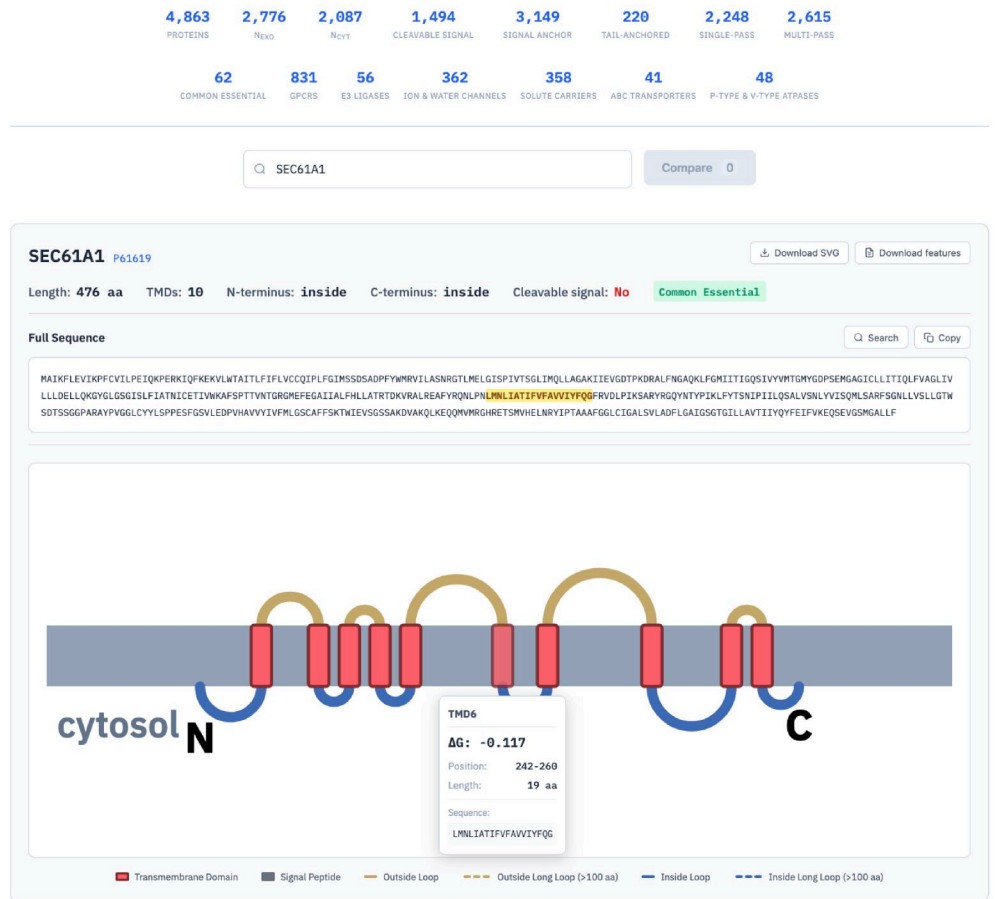

Figure S2. **Topology viewer for the human membrane proteome.** Screenshot of the topology viewer web interface, illustrated with SEC61A1 (UniProt P61619). The full protein sequence is displayed at top with the selected TMD highlighted in yellow. Hovering over any aa in the full sequence will highlight the corresponding segment in the topology diagram. The topology diagram depicts TMDs as red rectangles embedded in a gray membrane, with outside loops shown in gold and inside loops in blue; dashed lines indicate loops >100 aa. Hovering over any TMD, loop, tail, or SS displays a tooltip with their biophysical properties including $\Delta G_{app}$, position, length, location, and sequence (illustrated here for TMD6: $\Delta G_{app}$ = −0.117, position 242–260, 19 aa). The protein is flagged as a common essential gene. Specific sequences or motifs can be identified via a search function. For example, glycosylation sequons can be searched as NX(ST) or N!P(ST), where X indicates any aa, and !P excludes Pro at the second position, and include both Ser and Thr at the third position. The viewer also supports SVG diagram download and feature table download.

**Provided online is Table S1. Table S1 contains the topologic parameters for all membrane proteins analyzed in this study.**

