## [Peer Review File · The Journal of Cell Biology]

Design principles of human membrane protein topology

Haoxi Wu and Ramanujan Hegde

Corresponding Author(s): Haoxi Wu, University of Oxford and Ramanujan Hegde, MRC Laboratory of Molecular Biology

Review Timeline:	Submission Date:	2026-04-09
	Editorial Decision:	2026-05-11
	Revision Received:	2026-05-13

Monitoring Editor: Elizabeth Miller

Scientific Editor: Andrea Marat

Transaction Report:

DOI: <https://doi.org/10.1083/jcb.202604059>

May 11, 2026

RE: JCB Manuscript #202604059

Haoxi Wu
University of Oxford

Dear Dr. Wu:

Thank you for submitting your manuscript entitled "Design principles of human membrane protein topology". You will see that the three expert reviewers are very positive that your paper provides a highly useful resource for the community. They have some relatively minor concerns which can be addressed with text edits and clarifications. Therefore, we would be happy to publish your paper in JCB pending addressing their comments and final revisions necessary to meet our formatting guidelines (see details below).

A. MANUSCRIPT ORGANIZATION AND FORMATTING:

- 1) Text limits: Character count for Tools is < 40,000, not including spaces. Count includes abstract, introduction, results, discussion, and acknowledgments. Count does not include title page, figure legends, materials and methods, references, tables, or supplemental legends.
- 2) Figures limits: Tools may have up to 10 main text figures.
- 3) Figure formatting: Scale bars must be present on all microscopy images, including inset magnifications. Molecular weight or nucleic acid size markers must be included on all gel electrophoresis. Aspect ratios of images may not be altered.
- 4) Statistical analysis: Error bars on graphic representations of numerical data must be clearly described in the figure legend. The number of independent data points (n) represented in a graph must be indicated in the legend. Statistical methods should be explained in full in the materials and methods. For figures presenting pooled data the statistical measure should be defined in the figure legends. Please also be sure to indicate the statistical tests used in each of your experiments (either in the figure legend itself or in a separate methods section) as well as the parameters of the test (for example, if you ran a t-test, please indicate if it was one- or two-sided, etc.). Also, if you used parametric tests, please indicate if the data distribution was tested for normality (and if so, how). If not, you must state something to the effect that "Data distribution was assumed to be normal but this was not formally tested."
- 5) Abstract and title: The abstract should be no longer than 160 words and should communicate the significance of the paper for a general audience. The title should be less than 100 characters including spaces. Make the title concise but accessible to a general readership.
- 6) Materials and methods: Should be comprehensive and not simply reference a previous publication for details on how an experiment was performed. Please provide full descriptions in the text for readers who may not have access to referenced manuscripts.
- 7) All antibodies, cell lines, animals, and tools used in the manuscript should be described in full, including accession numbers for materials available in a public repository such as the Resource Identification Portal. Please be sure to provide the sequences for all of your primers/oligos and RNAi constructs in the materials and methods. You must also indicate in the methods the source, species, and catalog numbers (where appropriate) for all of your antibodies. Please also indicate the acquisition and quantification methods for immunoblotting/western blots.
- 8) Microscope image acquisition: The following information must be provided about the acquisition and processing of images:
 - a. Make and model of microscope
 - b. Type, magnification, and numerical aperture of the objective lenses
 - c. Temperature
 - d. Imaging medium
 - e. Fluorochromes
 - f. Camera make and model
 - g. Acquisition software

h. Any software used for image processing subsequent to data acquisition. Please include details and types of operations involved (e.g., type of deconvolution, 3D reconstitutions, surface or volume rendering, gamma adjustments, etc.).

10) Supplemental materials: There are strict limits on the allowable amount of supplemental data. Tools may have up to 5 supplemental figures. Please also note that tables, like figures, should be provided as individual, editable files. A summary of all supplemental material should appear at the end of the Materials and methods section.

13) ORCID IDs: ORCID IDs are unique identifiers allowing researchers to create a record of their various scholarly contributions in a single place. Please note that ORCID IDs are now *required* for all authors. At resubmission of your final files, please be sure to provide your ORCID ID and those of all co-authors.

Please note that JCB now requires authors to submit Source Data used to generate figures containing gels and Western blots with all revised manuscripts. This Source Data consists of fully uncropped and unprocessed images for each gel/blot displayed in the main and supplemental figures. For assays performed using capillary electrophoresis and/or immunoassay-based detection, authors should instead provide the electropherogram graph(s) for each experiment, plotting fluorescence/chemiluminescence intensity vs. molecular weight/size. Please be sure to provide one Source Data file for each figure gels, blots, and/or capillary electrophoresis assays along with your revised manuscript files. File names for Source Data figures should be alphanumeric without any spaces or special characters (i.e., SourceDataF#, where F# refers to the associated main figure number or SourceDataFS# for those associated with Supplementary figures). For traditional gels and blots, the lanes of the gels/blots should be labeled as they are in the associated figure, the place where cropping was applied should be marked (with a box), and molecular weight/size standards should be labeled wherever possible. For capillary electrophoresis assays, each trace in the graph should be color-coded and labeled to indicate which protein, gene, or sample is being measured (please try to avoid red/green combinations to accommodate our color-blind readers).

Journal of Cell Biology now requires a data availability statement for all research article submissions. These statements will be published in the article directly above the Acknowledgments. The statement should address all data underlying the research presented in the manuscript. Please visit the JCB instructions for authors for guidelines and examples of statements at (<https://rupress.org/jcb/pages/editorial-policies#data-availability-statement>).

B. FINAL FILES:

-- Cover images: If you have any striking images related to this story, we would be happy to consider them for inclusion on the journal cover. Submitted images may also be chosen for highlighting on the journal table of contents or JCB homepage carousel.

Images should be uploaded as TIFF or EPS files and must be at least 300 dpi resolution.

****It is JCB policy that if requested, original data images must be made available to the editors. Failure to provide original images upon request will result in unavoidable delays in publication. Please ensure that you have access to all original data images prior to final submission.****

****The license to publish form must be signed before your manuscript can be sent to production. A link to the license to publish form will be sent to the corresponding author only. Please take a moment to check your funder requirements before choosing the appropriate license.****

Thank you for your attention to these final processing requirements. Please revise and format the manuscript and upload materials within 14 days. If you need an extension for whatever reason, please let us know and we can work with you to determine a suitable revision period.

Thank you for this interesting contribution, we look forward to publishing your paper in Journal of Cell Biology.

Sincerely,

Elizabeth Miller, PhD
Monitoring Editor

Andrea L. Marat, PhD
Deputy Editor

Journal of Cell Biology

Reviewer #1 (Comments to the Authors (Required)):

In this Tool manuscript from Wu and Hegde, the authors have analyzed the human ER/secretory membrane proteome to generate a detailed topological "census", and have created a webserver for researchers to visualize the topology of human membrane proteins of interest. (Mitochondrial membrane proteins are not included in this analysis)

To generate this database, the authors "combine classical topology prediction algorithms with recent genome-wide protein structure predictions, together with manual inspection, to generate a census of human membrane protein topology."

The authors then perform an in-depth analysis of their database by analyzing amino acid compositions and biophysical properties. This leads them to highlight several findings of broad interest and significance including:

-The exoplasmic domain of multipass proteins tends to be much smaller than that of single-pass proteins.

-Type III (Nexo) multipass proteins usually have an odd-number of TMDs, whereas Type II (Ncyt) multipass proteins usually have an even number of TMDs. In both cases the result is that the C-terminus is in the cytosol.

-While the TMDs of most single-pass proteins are quite hydrophobic, TMDs of multipass proteins often contain charged residues and most do not appear to be hydrophobic enough for insertion by Sec61. Translocation of multipass proteins is thought to occur via a mechanism in which two TMDs are translocated together, and the authors find that when considering pairs of TMDs, the charge bias for each pair is similar to that seen for single-pass N-exo signal-anchor proteins.

Finally, the authors are able to correlate the distinct hydrophobicity and charge bias properties of different groups of membrane proteins with their different insertion machineries and mechanisms. For example, strong hydrophobicity and a strong positive-inside bias favors insertion of a single TMD by Sec61, while lower hydrophobicity and the presence of a flanking negative (outside) charge favors insertion of a TMD pair by the EMC. TMD pairs that are inserted by the MPT appear to be even more hydrophilic than those inserted by the EMC. They summarize and reconcile their findings in Figure 9 with an illustration of the 'six core reactions' of membrane protein insertion. A striking finding (at least to me) is their observation that a majority of TMDs

are inserted (as pairs) via EMC or MPT.

Overall I think the authors have created a very valuable resource and have performed a thorough and enlightening analysis. I think this Tool will be of broad interest to the readers of JCB and the membrane biology community.

The only concern I have relates to the topology assignments for a significant fraction of the ~2000 signal-anchor proteins which required manual assessment by the authors, as these proteins lack a signal-peptide that unambiguously establishes inside vs. outside topology. In many cases these assignments were made by the authors based on experimental evidence or homology to proteins with experimental evidence, but it appears that a large number of the topology assignments were based solely on the positive-inside rule. In Figure 5 and later figures, the authors analyze the charge distribution of the TMDs in their resulting database and declare that the positive-inside rule is "universal". If I am understanding correctly, in making this statement the authors are essentially analyzing the results of their own selection criteria used for making the assessments in the first place. So, I worry that there might be a bit of circular reasoning here, because they are using the same criteria for assignment and assessment?

Minor points:

Why does the number of single-pass Nexo proteins in Figure 6 (1282) not appear to be equal to the Nexo quantities in Figure 1?

The webserver will be quite helpful because as the authors point out, knowing the topology of membrane proteins is a quick and easy way to exclude certain hypotheses, such as computationally predicted interactions between domains that cannot interact with each other because they are on opposite sides of a membrane. I therefore wonder if it might be possible for the authors to assign a confidence metric to their topology assignment for each protein? If researchers are to use the webserver in designing their own experiments, it might be helpful for users to know how confident they should be in the predicted topology of their favorite protein(s)?

Reviewer #2 (Comments to the Authors (Required)):

This is a carefully executed and conceptually satisfying study combining rigorous manual curation with large scale structural and biophysical analysis to bring clarity to how eukaryotic membrane proteins (with humans as the exemplar) are organised and inserted. The authors' treatment of the ER membrane proteome is thorough and judicious, resolving many long standing annotation ambiguities and delivering a dataset that will be of great value to the community. What elevates the work beyond a resource paper is the identification of transmembrane domain pairs as the fundamental architectural and biogenetic unit of multipass proteins, coupled to a persuasive mechanistic argument that such pairs are handled primarily by Oxa1 family insertases, rather than the Sec61/Y family. The analyses are well controlled, and the conclusions firmly grounded-consistent with our understanding of those translocons. The dissection of complex behaviour to a small number of core insertion reactions is particularly effective. Overall, this is a substantial contribution that advances both our mechanistic understanding of membrane protein biogenesis, and it merits publication as is.

Reviewer #3 (Comments to the Authors (Required)):

Wu and Hegde have conducted an in-depth sequence-based analysis of ~5,000 human membrane proteins, conditioned on recently acquired knowledge of the of membrane-insertion mechanisms followed by different types of membrane proteins. Statistical studies of this kind go back at least ~40 years, and many of the results reported in this study confirm what is already known. For the most part the authors acknowledge this (even if they do not always include the salient references, see below). The one case where they do not, in my opinion, correctly reflect previous knowledge is in their discussion of "TMD-pairs" (or "helical hairpins") as being important structural elements in the membrane insertion of multi-spanning proteins - this idea is not novel (though one can read the Ms as if it is) and goes back some 45 years. Likewise, I think that it's a bit misleading to state that "the first TMD has long been thought to be critical for setting overall topology" (p 8) - it is true that this idea was entertained in the early days of the field, but has long been abandoned (because of ample experimental and bioinformatics evidence to the contrary).

This having been said, the Ms does provide an interesting perspective by interpreting the statistical results in the context of the recent, integrated model of membrane protein biogenesis pathways, as depicted in Fig. 9.

Technically, the study is competently done. There is no formal statistical analysis of the results, but, given the large number of sequences included, even small differences between different data sets will be statistically significant.

Specific comments

- Section "Topologic features of the membrane proteome": Essentially all of the reported results confirm earlier studies (even if the precise numbers differ a bit), though no references are given. Also, the statement on the distance from the TMDs of N-linked glycosylation sequons should be backed up by a reference.

- Section "Multipass proteins are mostly built of TMD-pairs": Essentially all of the reported results confirm earlier studies - including the results on GPCRs (e.g., Prot Sci 7:693) - (even if the precise numbers differ a bit), though no references are given. This also holds for the concept of "TMD-pairs" (see above).

- Section "Composition and properties of TMDs": Essentially all of the reported results confirm earlier studies (even if the precise numbers differ a bit). A couple of references are given, so that's OK.

- Section "Charged amino acids are prevalent in multipass proteins": Essentially all of the reported results confirm earlier studies (even if the precise numbers differ a bit). A couple of references are given, so that's OK.

- Section "The positive-inside rule is weaker for individual TMDs of multipass proteins": The results are in concert with earlier results (though now more detailed). The small negative-outside bias has been noted before (originally for a much smaller dataset; Prot Sci 7:1029). See the comment above on the relative importance of the first TMD. I don't think that the charge bias and summed hydrophobicity have been calculated for TMD-pairs before, so this is new.

- Sections "Signal anchor properties reflect their different insertion mechanisms", "A preceding TMD facilitates insertion of the next TMD through Sec61", and "High TMD hydrophilicity is tolerated in TMD-pairs of multipass proteins": Such a detailed level of analysis has not been published before, to my knowledge.

- Section "The membrane proteome is produced by six core insertion reactions": A nice summary of the results interpreted in the context of the recent model for the mechanisms followed by different types of membrane proteins developed by the Hedge lab.

In summary, the dataset is organized in a way that allows a user to easily identify the particular membrane insertion pathway used by a particular protein. The analysis presented in the paper provides an interesting overview of the sequence features that correlate with the different insertion pathways and interprets these features in light of current knowledge about the structure and function of the different translocons involved. However, previous work could be more comprehensively cited.

Response to referee comments:

We thank all reviewers for their positive and constructive feedback on our manuscript. We have addressed each of the comments below.

Reviewer #1:

The only concern I have relates to the topology assignments for a significant fraction of the ~2000 signal-anchor proteins which required manual assessment by the authors, as these proteins lack a signal-peptide that unambiguously establishes inside vs. outside topology. In many cases these assignments were made by the authors based on experimental evidence or homology to proteins with experimental evidence, but it appears that a large number of the topology assignments were based solely on the positive-inside rule. In Figure 5 and later figures, the authors analyze the charge distribution of the TMDs in their resulting database and declare that the positive-inside rule is "universal". If I am understanding correctly, in making this statement the authors are essentially analyzing the results of their own selection criteria used for making the assessments in the first place. So, I worry that there might be a bit of circular reasoning here, because they are using the same criteria for assignment and assessment?

This is a good point. We now note on pg. 8 that similar results are seen for the charge distribution of signal-anchors from proteins of well-studied families whose topology can be unambiguously assigned independently of charge. For 166 single-pass proteins (cytochrome P450s, glycosyltransferases, and E3 ligases), the average net exoplasmic charge is -0.13 and net cytosolic charge is +2.51. For multipass proteins (GPCRs, E3 ligases, ABC transporters, channels, solute carriers, and ATPases), the average net exoplasmic charge is -0.12 and net cytosolic charge is +1.16. This is consistent with the values reported in Figure 5 for all single-pass and all multipass proteins.

Minor points:

Why does the number of single-pass Nexo proteins in Figure 6 (1282) not appear to be equal to the Nexo quantities in Figure 1?

This seems to be a misunderstanding. The 1282 number in Fig. 6A is for all N_{exo} signal anchors, not just the single-pass ones. Fig. 6B splits these 1282 proteins between single-pass (304) and multipass (978), which are the same numbers as in Fig. 1.

The webserver will be quite helpful because as the authors point out, knowing the topology of membrane proteins is a quick and easy way to exclude certain hypotheses, such as computationally predicted interactions between domains that cannot interact with each other because they are on opposite sides of a membrane. I therefore wonder if it might be possible for the authors to assign a confidence metric to their topology assignment for each protein? If researchers are to use the webserver in designing their own experiments, it might be helpful for users to know how confident they should be in the predicted topology of their favorite protein(s)?

We agree that this would be really great. Unfortunately, we have not been able to devise an unbiased strategy given the multiple types of evidence underlying our assignments (ranging from experimental structural data to structure prediction to homology to simple sequence analysis). One option we are considering is to tabulate the type(s) of evidence used in assigning topology, but we struggle with weighting them appropriately to avoid a misleading metric. We will continue to work on this and hope to implement an update in the future.

Reviewer #2: No comments to address.

Reviewer #3:

Wu and Hegde have conducted an in-depth sequence-based analysis of ~5,000 human membrane proteins, conditioned on recently acquired knowledge of the of membrane-insertion mechanisms followed by different types of membrane proteins. Statistical studies of this kind go back at least ~40 years, and many of the results reported in this study confirm what is already known. For the most part the authors acknowledge this (even if they do not always include the salient references, see below). The one case where they do not, in my opinion, correctly reflect previous knowledge is in their discussion of "TMD-pairs" (or "helical hairpins") as being important structural elements in the membrane insertion of multi-spanning proteins - this idea is not novel (though one can read the Ms as if it is) and goes back some 45 years.

This is an embarrassing oversight, and we apologize. We are of course well aware of (and have cited in our earlier papers) the pioneering work on the "helical hairpin" hypothesis of Engelman and Steitz (1981) and Steitz et al. (1982). This is also consistent with the many genome-wide observations and findings in the seminal work by Wallin and von Heijne (1998). These citations are now included at key relevant points in the revised manuscript.

Likewise, I think that it's a bit misleading to state that "the first TMD has long been thought to be critical for setting overall topology" (p 8) - it is true that this idea was entertained in the early days of the field, but has long been abandoned (because of ample experimental and bioinformatics evidence to the contrary).

We agree, although it is fair to say that outside the field, the original idea remains stubbornly pervasive. We have therefore changed "has long been thought" to "was initially thought and still frequently cited" to be critical for setting overall topology (pg. 8).

Specific comments

- Section "Topologic features of the membrane proteome": Essentially all of the reported results confirm earlier studies (even if the precise numbers differ a bit), though no references are given. Also, the statement on the distance from the TMDs of N-linked glycosylation sequons should be backed up by a reference.

We agree. We now cite Wallin and von Heijne (1998) to reflect similar earlier genome-wide analyses and observations. We have added a citation to Nilsson and von Heijne (1993) for the minimum sequon-to-TMD distance required for N-linked glycosylation.

- Section "Multipass proteins are mostly built of TMD-pairs": Essentially all of the reported results confirm earlier studies - including the results on GPCRs (e.g., Prot Sci 7:693) -(even if the precise numbers differ a bit), though no references are given. This also holds for the concept of "TMD-pairs" (see above).

We have added citations to Wallin and von Heijne (1995) for the GPCR work, to Wallin and von Heijne (1998) for the observation of closely-spaced TMDs, and to the Engelman and Steitz papers for the hypothesis of TMD-pair insertion.

- Section "The positive-inside rule is weaker for individual TMDs of multipass proteins": The results are in concert with earlier results (though now more detailed). The small negative-outside bias has been noted before (originally for a much smaller dataset; Prot Sci 7:1029). See the comment above on the relative importance of the first TMD. I don't think that the charge bias and summed hydrophobicity have been calculated for TMD-pairs before, so this is new.

We have added a citation to Wallin and von Heijne (1998) where the weak negative-outside bias was previously noted.